# The Courage to Stop: Overcoming Sunk Cost Fallacy in Deep Reinforcement Learning

Jiashun Liu [1]   Johan Obando-Ceron [2 3]   Pablo Samuel Castro [2 3]   Aaron Courville [2 3]   Ling Pan [1]

## Abstract

Off-policy deep reinforcement learning (RL) typically leverages replay buffers for reusing past experiences during learning. This can help improve sample efficiency when the collected data is informative and aligned with the learning objectives; when that is not the case, it can have the effect of "polluting" the replay buffer with data which can exacerbate optimization challenges in addition to wasting environment interactions due to wasteful sampling. We argue that sampling these uninformative and wasteful transitions can be avoided by addressing the *sunk cost fallacy* which, in the context of deep RL, is the tendency towards continuing an episode until termination. To address this, we propose *learn to stop* (LEAST), a lightweight mechanism that enables strategic early episode termination based on $Q$-value and gradient statistics, which helps agents recognize when to terminate unproductive episodes early. We demonstrate that our method improves learning efficiency on a variety of RL algorithms, evaluated on both the MuJoCo and DeepMind Control Suite benchmarks.

*"People are reluctant to waste prior investments, even when continuing guarantees further loss."*

– Halr Arkes

## 1. Introduction

The *sunk cost fallacy* refers to a behavioral pattern where agents continue along a current course of action despite recognizing its suboptimality, rather than stop acting if they have already devoted a certain amount of effort towards it (Arkes & Blumer, 1985; Sweis et al., 2018). This cognitive bias often leads to inefficiencies in decision-making

[1]Hong Kong University of Science and Technology [2]Mila - Québec AI Institute [3]Université de Montréal. Correspondence to: Ling Pan <lingpan@ust.hk>.

*Proceedings of the 42nd International Conference on Machine Learning*, Vancouver, Canada. PMLR 267, 2025. Copyright 2025 by the author(s).

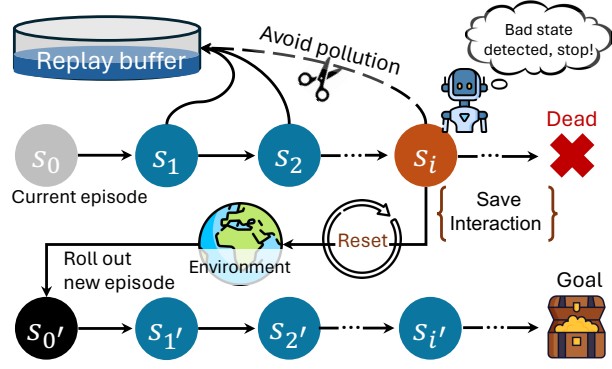

*Figure 1.* LEAST enables the agent to prematurely end the current episode by monitoring the quality of present situations, such as getting stuck in a suboptimal trajectory. This mechanism improves sample efficiency, reduces replay buffer contamination, and conserves the overall interaction budget.

processes. A simple example of this is watching a bad movie until the end because we have already paid for a movie ticket, thereby wasting our time and energy that could have been devoted elsewhere. In this work, we argue that this fallacy is inherent in traditional deep reinforcement learning (deep RL) frameworks, where enforced episode completion hinders agents from making strategic stopping decisions (Pardo et al., 2018). It gives the impression that the RL agent is fixated on the interaction costs already incurred, which limits its subsequent learning capability and sample efficiency.

Specifically, the design limitations of traditional RL, i.e., lacking key mechanisms for autonomous stopping decisions, can limit agents to repeatedly traverse low-quality (suboptimal) trajectories similar (uninformative) to those they have encountered several times before. Although utilizing experience replay buffers can improve sample efficiency for off-policy deep RL (Fedus et al., 2020), the accumulation of such experiences, particularly during early training stages, can require parameter-sensitive sampling methods (Schaul, 2015) to prevent overfitting to uninformative experiences (Nikishin et al., 2022).

Furthermore, this constrained decision space and restricted decision flexibility fail to optimize the environmental interaction budget (Yin et al., 2023). To address the aforemen-

tioned fundamental challenge, we introduce Learn to Stop (LEAST), a novel approach that empowers RL agents with the ability to terminate episodes prematurely, illustrated in Figure 1.

The agent learns to evaluate the trade-off between continuing interaction with the environment and devotes more interaction costs and potential benefits considering statistical information of historical experiences. The core of LEAST is an adaptive stopping mechanism that dynamically determines whether to terminate the current trajectory based on its quality and how well we have learned such experiences.

Specifically, we construct an adaptive stopping threshold by jointly analyzing two key aspects of the agents' past experiences, the central tendency of action values $Q$ and the gradient of critic, which reflect the quality and the learning potential of recent transitions (how well the agent has mastered the current trajectory), separately.

During environment interaction, LEAST compares the predicted $Q$-value $\hat{Q}$ against the adaptive threshold. When $\hat{Q}$ falls below the threshold, it provides an indication that the current partial trajectory's quality and similarity to well-learned experiences can be inferior to the historical statistics in memory, triggering a strategic early termination of the current episode.

We conduct extensive experiments in standard continuous control MuJoCo tasks and the challenging DeepMind Control Suite benchmarks to demonstrate its effectiveness. On all benchmark tasks, LEAST significantly outperforms the baseline algorithms. Our work opens up a novel direction by teaching agents when to "cut losses"[1], which is a fundamental capability in human decision-making and represents a novel and orthogonal approach to existing methods for improving efficiency. The main contributions of this paper are summarized as follows:

- We identify a critical limitation in current deep RL frameworks where agents continue interacting with the environment even when encountering familiar low-quality trajectories, leading to wasted environmental interactions and increased sample complexity.

- We propose LEAST: a simple yet effective mechanism that enables strategic stopping, significantly improving sample efficiency and performance of existing algorithms.

- We validate the effectiveness of LEAST through comprehensive experiments across standard benchmarks including MuJoCo (Brockman, 2016) and visual RL scenarios with DeepMind Control (Tassa et al., 2018). Our results demonstrate consistent and significant improvement, illustrating LEAST's efficiency and versatility.

---

[1]Please refer to Appendix 5 for detailed related work on sample efficiency and early stopping.

## 2. Background

**Deep RL**    The problem of reinforcement learning is commonly framed within the framework of a Markov decision process (MDP), which is defined by a tuple $(\mathcal{S}, \mathcal{A}, \mathcal{P}, \mathcal{R}, \gamma)$. Here, $\mathcal{S}$ represents the state space, $\mathcal{A}$ denotes the action space, $\mathcal{P}$ describes the transition dynamics as a function $\mathcal{S} \times \mathcal{A} \times \mathcal{S} \rightarrow [0, 1]$, $\mathcal{R}$ is the reward function defined as $\mathcal{S} \times \mathcal{A} \rightarrow \mathbb{R}$, and $\gamma$ (where $\gamma \in [0, 1)$) signifies the discount factor.

The agent interacts with the environment with its policy $\pi$, which maps states to actions. The goal is to acquire an optimal policy that maximizes the expected discounted long-term reward. The expected value of a state-action pair $(s, a)$ is defined as $Q^\pi(s, a) = \mathbb{E}\pi[\sum t = 0^\infty \gamma^t \mathcal{R}(s_t, a_t)|s_0 = s, a_0 = a]$, which means the sum of discounted rewards starting from current transition $(s, a)$.

In actor-critic methods, the actor $\pi_\phi$ and the critic $Q_\theta$ are typically implemented using neural networks, where the parameters of the actor and critic networks are denoted as $\phi$ and $\theta$, respectively. The critic network is updated by minimizing the temporal difference loss, i.e., $\mathcal{L}_Q(\theta) = \mathbb{E}_D\big[(Q_\theta(s, a) - Q_\theta^{\mathcal{T}}(s, a))^2\big]$, where $Q^{\mathcal{T}}(s, a)$ denotes the bootstrapping target $\mathcal{R}(s, a) + \gamma Q_{\bar{\theta}}(s', \pi_{\bar{\phi}}(s'))$ computed using target network parameterized by $\bar{\phi}$ and $\bar{\theta}$ based on data sampled from a replay buffer $D$. The actor network $\phi$ is typically updated to maximize the Q-function approximation according to $\nabla_\phi J(\phi) = \mathbb{E}_D\left[\nabla_a Q_\theta(s, a)|_{a=\pi_\phi(s)} \nabla_\phi \pi_\phi(s)\right]$.

**Sunk Cost Fallacy**    Sunk cost fallacy (Turpin et al., 2019), also known as sunk cost bias (Arkes & Blumer, 1985), is widely discussed in cognitive science. It describes the tendency to follow through with something that people have already invested heavily in (be it time, money, effort, or emotional energy), even when giving up is clearly a better idea. This phenomenon is particularly prevalent in the world of investments (Dijkstra & Hong, 2019). Whether it's in the stock market, real estate, or even a business venture, investors often fall into the trap of holding onto underperforming assets because they've already sunk so much money into them (Haita-Falah, 2017). In this paper, we investigate whether deep RL agents deployed on cumulative reward maximization tasks also suffer from sunk cost fallacy.

## 3. Learn to Stop (LEAST)

In this section, we initially investigate whether the sampling process within the current deep RL algorithm could induce the agent to exhibit signs of sunk cost fallacy and explore the impact of the sunk cost fallacy on learning efficiency, as discussed in §3.1. Subsequently, we introduce a straightforward and general optimization to the conventional algo-

rithmic framework in §3.2, aimed at enabling the agent to dynamically overcome sunk cost fallacy, thereby enhancing the learning efficiency of prevalent deep RL methods.

### 3.1. Motivating Example

The sunk cost fallacy, a prevalent cognitive bias where agents persist with their current course of action considering prior efforts (Dijkstra & Hong, 2019), is inherently present in current deep RL frameworks. Without the decision flexibility to terminate episodes strategically, agents using popular deep RL algorithms, like TD3 (Fujimoto et al., 2018) and SAC (Haarnoja et al., 2018), lack the capability to learn to stop, forcing them to complete entire episodes even when already traversing familiar low-quality trajectories. It leads to increased interaction costs, and can also result in the replay buffer becoming filled with suboptimal experiences.

This phenomenon aligns with findings in value-based deep RL, where using smaller batch sizes has been shown to improve performance by increasing gradient noise, which serves as a form of implicit regularization and mitigates overfitting to low-quality transitions in the replay buffer (Ceron et al., 2023). To quantify the impact of this limitation, we conduct controlled experiments to assess the performance disparity between vanilla '*fully executed*' deep RL agent and an advanced agent, i.e., our method LEAST, equipped with auto-stopping capability (which learns to stop upon entering a low-quality trajectory that it is already familiar). LEAST is introduced in §3.2.

We implement both agents based on Soft Actor-Critic (SAC) (Haarnoja et al., 2018) as the backbone algorithm, and evaluate its learning efficiency on a challenging Point-Maze benchmark as detailed introduced in Appendix A.2 from Hu et al. (2023), where there are a number of lengthy branches that can trap agents in unproductive trajectories. To ensure fairness, both agents are trained for $10^6$ timesteps with identical hyperparameters and a replay buffer size of $10^5$. The statistics of the trajectories are visualized in Appendix A.2 which shows that the vanilla agent spends too much time in the "unproductive trajectories".

Figure 2 demonstrates significantly higher learning efficiency of the advanced agent with auto-stopping capability compared to the vanilla agent. As the horizon of the maze expands, interaction budgets become more constrained, leading to a correspondingly greater demand for sample efficiency. The vanilla agent's performance is significantly reduced (maintaining at 60 score), which proves that it is trapped in suboptimal trajectories and unable to escape, and the optimization speed is slow due to the decreased sample efficiency. However, the advanced agent nearly maintains a stable performance by avoiding wasteful exploration of known low-quality transitions. This advantage becomes even more pronounced in long-horizon tasks.

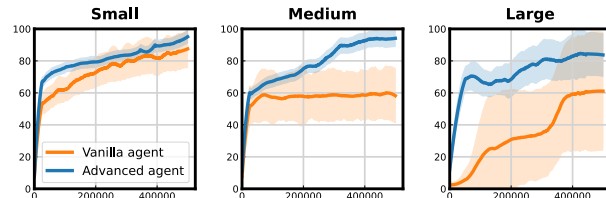

*Figure 2.* **Comparison between the vanilla agent and our proposed LEAST agent across three PointMaze environments with identical layouts but varying horizon lengths (10, 18, 24).** Results are averaged over 5 seeds. The Y-axis denotes the normalized score (out of 100), while the X-axis indicates training timesteps. Task details are provided in Appendix A.2.

Finally, we analyze the data distribution within the agents' replay buffer (Figure 3). We partition the data into quadrants based on the mean values of $Q$ (expected reward) and loss (learning signal) from the advanced agent's buffer samples. The advanced agent significantly reduces the proportion of uninformative samples in the buffer, specifically, those with both low $Q$-values and low loss (shown in white), which offer little to no effective learning signal. This is achieved by terminating low quality trajectories early. As a result, the buffer contains a higher proportion of high quality data (shown in black), enabling the agent to sample more informative experiences during training and thereby improving learning efficiency.

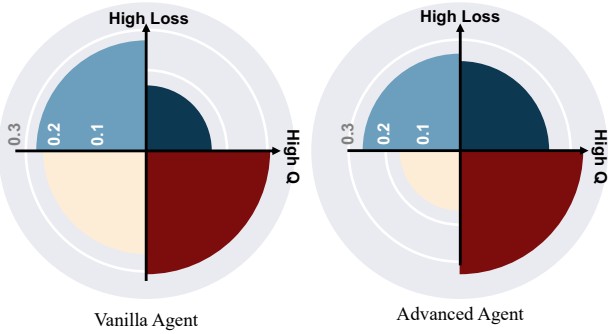

*Figure 3.* **Replay buffer data distribution at $2 \times 10^5$ timesteps.** Compared to the vanilla agent, the advanced agent with LEAST significantly reduces the proportion of low-quality transitions (white region: low $Q$-value and low learning signal) and increases the density of high-quality samples (black region: high $Q$-value and high gradient magnitude), resulting in a more informative and efficient training dataset.

### 3.2. Proposed Method

In §3.1, we observed that agents could mitigate the impact of the sunk cost fallacy by stopping and resetting before descending into suboptimal behaviors. In this section, we propose a lightweight general mechanism *Learn to Stop*

(LEAST) to unlock the agent's auto-stop ability. The high-level idea is to use the recent central tendency of behavior to construct a sensitivity-aware stop threshold for the current execution. Based on this adaptive threshold, agents can evaluate the risk of falling into suboptimality and decide whether to stop. Empirically, we find that two key factors are essential for the effectiveness of LEAST: (i) identifying the quantifiable and common elemental data to systematically characterize recent policies, and (ii) statistics features of dynamic time scales to sensitively fit on-changing policies.

**Autonomous Stopping Mechanism** The key challenge in designing an effective stopping mechanism is to evaluate the quality and potential learning value of ongoing trajectories during agent-environment interaction, and making timely stopping decisions to save interaction costs to redirect the limited interaction budget to more promising experiences for improving sample efficiency. To address these challenges, we propose a dual-criteria approach that dynamically evaluates current trajectories from complementary perspectives including the quality and learning potential.

We leverage $Q$-values as the primary quality metric for determining whether an ongoing trajectory is worth continuing.[2] This choice is motivated by its property that they quantify the expected cumulative rewards for state-action pairs and capture the long-term values of decisions (Van Hasselt et al., 2016). Specifically, we maintain a two-dimensional buffer $\mathcal{B}_Q \in \mathbb{R}^{K \times L}$ that stores $Q$-values from the $K$ most recent episodes ($L$ indicates the maximum episode length), which provides a dynamic reference for quality assessment. Then, we compare the current $Q$-value $\hat{Q}(s_i, a_i)$ at step $i$ with a threshold $\epsilon$ derived from historical experience.

When $\hat{Q}_i < \epsilon$, it indicates that the expected return of the ongoing trajectory falls below the historical average, suggesting potentially suboptimal behavior that warrants early termination. However, different stages of an episode have inherently different $Q$-value scales and behavioral requirements. We therefore consider step-based independent thresholds $\{\epsilon_1, ..., \epsilon_l\}$ for each step instead of a single universal threshold throughout the episode.[3] Considering the noisy distribution of $Q$-values particularly in the early stages of training, we use the median instead of the arithmetic mean for a more robust threshold calculation against outliers. Formally, for each intra-episode step $i$, the quality-based criterion is defined as:

$$\text{LEAST} := \text{Stop \& Reset if } \hat{Q}_i < Median(\mathcal{B}_Q[:, i]) \quad (1)$$

The effectiveness of the design choice is empirically validated in Figure 4, where the comparison between the orange

[2]Note that our method is built upon clipped double Q-learning that mitigates overestimation bias of $Q$-values.

[3]We use each minimum value of the subsequent $g$ step-axis after the stop step ($\min(\mathcal{B}_Q[:, g :])$) to fill the empty positions.

and brown boxes demonstrates the improved stopping behavior derived by our method.

While $Q$-values provide a measure of expected returns for quality, relying solely on them for stopping decisions overlooks a crucial aspect of RL that learns through trial and error: the value of exploration and learning from informative and novel experiences (Pathak et al., 2017; Rashid et al., 2020). Even trajectories with temporarily lower $Q$-values can provide valuable learning opportunities if they explore unfamiliar state-action regions.

Inspired by Sujit et al. (2023); Terven et al. (2023), we leverage the magnitude of the gradient of the $Q$-function as a proxy for learning potential, without introducing additional training module and use readily available information from the $Q$-function.

The intuition is that larger gradients indicate that the agent is encountering novel or poorly understood situations that merit further exploration. We maintain a parallel gradient buffer $\mathcal{B}_G$ (with the same structure as $\mathcal{B}_Q$), storing the gradient magnitudes from recent episodes. For each step $i$, we compute a dynamic weight $\omega_i$ as

$$\omega_i = \frac{Median(\mathcal{B}_G[:, i])}{G_i}, \quad (2)$$

where $G_i$ is the current gradient magnitude at step $i$.

To jointly consider quality and learning potential, we modulate our original quality-based threshold $\epsilon_i$ with the learning potential weight $\omega_i$ according to $\omega_i \times \epsilon_i$. Consider the case with a positive $\epsilon_i$, $\omega_i < 1$ indicates high exploration and learning potential for unfamiliar states (current gradient $G_i$ exceeds the historical median), which leads to a reduced threshold that encourages continued interaction. On the contrary, $\omega_i > 1$ will lead to stricter stopping criteria. Our complete stopping criterion in LEAST for each intra-episode step $i$ is formalized as follows:

$$\epsilon_i \geq 0 : \text{LEAST} := \text{Stop \& Reset if } \hat{Q}_i < \omega_i \times \epsilon_i$$
$$\epsilon_i < 0 : \text{LEAST} := \text{Stop \& Reset if } \hat{Q}_i < \omega_i^{-1} \times \epsilon_i. \quad (3)$$

When $\epsilon_i < 0$, using $\omega$ will lead to the opposite effect, for which we switch to $\omega^{-1}$. Figure 4 shows this dual-criteria threshold ($\omega_i \times \epsilon_i$) significantly outperforms purely $Q$-based threshold and further improves the performance by effectively trading off quality with potential learning value.

**Entropy-Aware Dynamic Buffer Size** An unstable policy may cause adjacent collected trajectories to vary significantly, thereby increasing the distributional uncertainty within $\mathcal{B}_Q$ and making it difficult to estimate its central tendency. To address this, we compute the entropy $H(\mathcal{B}_Q) = -\sum P(\mathcal{B}_Q) \log P(\mathcal{B}_Q)$ every $c$ training steps to assess the orderliness of $\mathcal{B}_Q$, and dynamically adjust the size

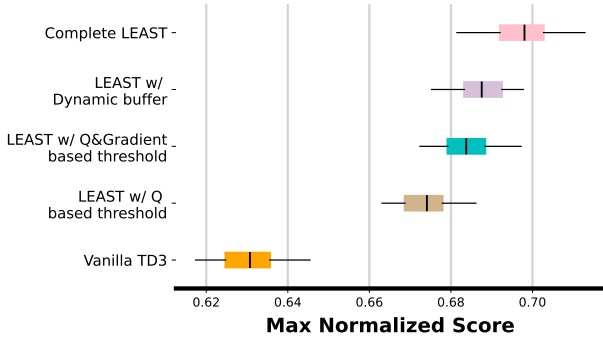

*Figure 4.* **Validation of modules, normalized performance on a high-dim task**, i.e., large maze (10 seeds). The box records the final scores of all seeds, the black line inside the box represents the median, while the horizontal line outside the box records the maximum and minimum values.

of the local buffer $\mathcal{B}_{Q,L}$. If the current entropy $H_t$ exceeds $(1 + \gamma) \times \bar{H}$, where $\bar{H}$ is a predefined baseline, we add (or remove) $h$ recent episodes from $\mathcal{B}_{Q,L}$ accordingly.

---

**Algorithm 1** TD3 with LEAST

1: Actor: $\pi_\phi$. Double critic networks: $Q_{\theta_{\{1,2\}}}$. Start timestep of LEAST: $t_{start}$. TD loss: $L$. Estimated Q value: $\hat{Q}$. Reflection Sets: $\mathcal{B}_Q, \mathcal{B}_G$. Dynamic weight: $\omega$. Stop threshold: $\epsilon$. Replay buffer: $D$. pre-set entropy of $\mathcal{B}_Q$: $\bar{H}$. Current entropy of $\mathcal{B}_Q$: $H_t$. Maximum overflow rate: $\gamma$, Adjusting scale: $h$.
2: **while** $t <$ maximum training time **do**
3:     **for** $i$ in $range(fixed\ episode\ length)$ **do**
4:         $a \leftarrow \pi_\phi(s)$ (+ noise schedule){Eq.4}
5:         Observe $r$ and new state $s'$
6:         # Update buffers
7:         Fill $D$ with $(s, a, r, s')$
8:         $\hat{Q}_1, \hat{Q}_2 \leftarrow Q_{\theta_{\{1,2\}}}(s,a)$; $L_i \leftarrow$ TD error
9:         Fill $\mathcal{B}_Q$ with $\min(\hat{Q}_1, \hat{Q}_2)$; Fill $\mathcal{B}_G$ with $L_i$
10:       # Entropy-based dynamic reflection set
11:       **if** $H_t > (1 + \gamma) \times \bar{H}$ **then**
12:         Add (remove) $h$ episodes $\rightarrow \mathcal{B}_{Q,L}${§3.2}
13:       # Start LEAST
14:       **if** $t \geq t_{start}$ **then**
15:         $\omega_i \leftarrow \frac{Median(\mathcal{B}_G[:,i])}{L_i}$
16:         $\epsilon_i \leftarrow Median(\mathcal{B}_Q[:,i])$
17:         Stop&Reset or not $\leftarrow$
            LEAST$(\omega_i, \epsilon_i)${Eq.3}
18:     Update $Q_{\theta_{\{1,2\}}}, \pi_\phi$ via Fujimoto et al. (2018)

---

**Adjusting Exploration Noise**    Empirically, we find that on individual tasks, e.g. Ant, agents have a probability of falling into the familiar suboptimal trajectory after stop & reset due to the unchanged policy, which may limit the benefits of adaptive stopping. However, the cumulative $Q$ estimating

error in algorithms makes the idea of accelerating behavior change by increasing the UTD affect the performance (Related experiments located in Appendix C.2).

Thus, inspired by Xu et al. (2023), we introduce a general episode-level schedule for regulating the exploration noise: *if agents are forced to stop at an early stage frequently, which means the policy may stuck in a suboptimal landscape, we will increase the exploration noise $\sigma$ (standard deviation of the Gaussian noise) to help them escape from current behavior.* Equation 4 defines how noise is adaptively increased when early stopping occurs frequently:

$$\sigma = \max\left(\frac{\bar{\sigma}}{1 + \exp(-\beta \times \tau + \mu)}, \sigma*\right), \sigma \in [\sigma*, \bar{\sigma}), \quad (4)$$

$\beta$ is the frequency of the most recent $m$ episodes to stop before the $e$-th step, where $\bar{\sigma}$ denotes the upper bound of the noise and $\sigma*$ is the original setting (lower bond), $\tau, \mu$ are the pre-set temperatures to make sure $\sigma$ can decay to $\sigma*$. A visualization of the noise schedule is shown in Figure 5.

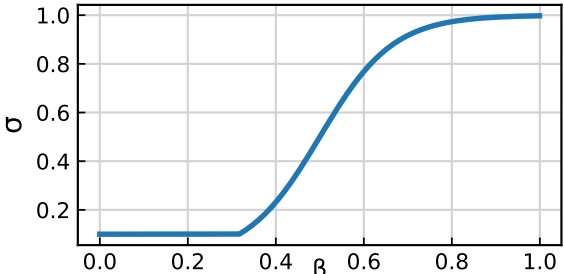

*Figure 5.* **Schedule of exploration noise adjustment.** As the frequency of premature stopping increases, indicating potential stagnation in policy improvement, the standard deviation of the exploration noise is gradually increased to encourage behavioral diversification and escape from suboptimal trajectories.

The curve of our complete method in Figure 4 shows that the performance of LEAST becomes more stable with a dynamic buffer size and noise schedule.

**Practical Implementation**    Pseudocode 1 illustrates the implementation of LEAST integrated with TD3, serving as a representative example of how our method can be applied to deterministic policy gradient algorithms.

## 4. Experiments

In this section, we conduct comprehensive experiments to evaluate whether LEAST module is necessary for deep RL. We investigate the following key questions: (**i**(§4.1)) Can LEAST improve the learning performance of mainstream deep RL algorithms? (**ii**(§4.2)) Is learning to stop also effective in more complex, image-based visual RL tasks?

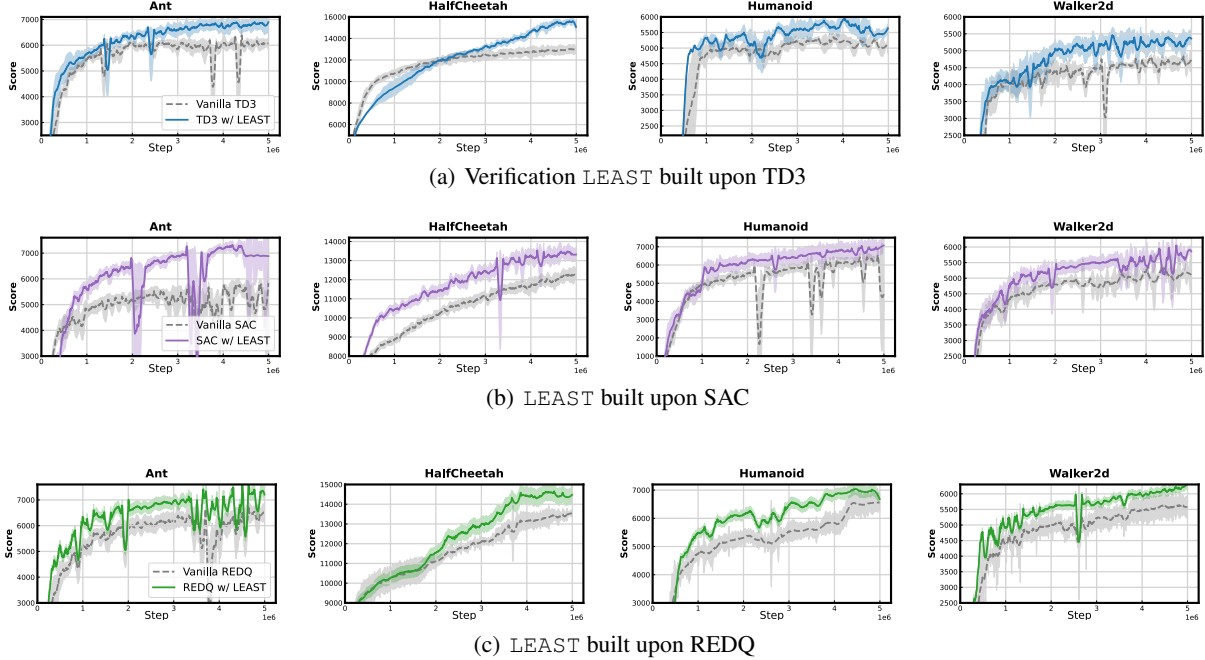

*Figure 6.* **Empirical validation of LEAST across diverse deep RL algorithms in MuJoCo environments.** Results show mean ± standard deviation over 5 runs. LEAST consistently improves both sample efficiency and final performance across: (a) TD3, representing deterministic policy methods; (b) SAC, based on maximum entropy reinforcement learning; and (c) REDQ, which employs an ensemble of critics for improved value estimation.

**iii**(§4.3)) How sensitive is LEAST to key parameters in each module, as revealed by detailed ablation studies?

### 4.1. Improvements for mainstream algorithms

**Experimental Setup** To verify the generality of LEAST, we select several widely used and mature deep RL algorithms as baselines: (i) TD3 (Fujimoto et al., 2018), representing deterministic policy methods; (ii) SAC (Haarnoja et al., 2018), as a representative of stochastic policies; (iii) REDQ (Chen et al., 2021), a sample-efficient method that uses an ensemble of critics to improve $Q$-function fitting; and (iv) DroQ (Hiraoka et al., 2021), a recent variant of SAC that incorporates dropout regularization.

We evaluate all methods on four core tasks from OpenAI Mujoco-v4, specifically, tasks that SAC and TD3 fail to solve effectively, serving as challenging benchmarks. To ensure fair comparison, we implement all methods using the official codebases and apply consistent hyperparameters across experiments. Detailed parameter settings are provided in Appendix B.

**Results** Overall, Figure 6 shows that LEAST significantly improves both learning efficiency and final performance across all three categories of deep RL algorithms, demonstrating its general effectiveness for mainstream deep RL. Specifically, for TD3 and SAC (Figure 6(a, b)), LEAST ac-

celerates early-stage learning and helps the agent converge to better policies, with particularly notable gains in the Ant and Walker2d tasks. Interestingly, while TD3 with LEAST achieves higher final scores in HalfCheetah, its convergence is slower in the early phase. This may be due to vanilla TD3 prematurely converging to suboptimal strategies. LEAST mitigates this by promoting more effective exploration, enabling the agent to eventually discover better behaviors in complex decision spaces.

In the case of REDQ (Figure 6(c)), since REDQ already enhances Q-function fitting via additional critics, its baseline performance is relatively strong. As a result, the performance gains from LEAST are less pronounced during early training. However, LEAST improves sample efficiency during the middle and later stages and further optimizes final performance.

Figure 7 provides a visual summary of sample efficiency improvements. The results indicate that LEAST significantly accelerates learning across all three algorithms, with particularly clear gains for TD3 compared to the stochastic-policy baselines. Finally, Figure 8 presents normalized scores across the four tasks to quantify the overall improvements. With LEAST, all algorithms show substantial performance gains: SAC with LEAST reaches performance comparable to DroQ, and REDQ with LEAST achieves the highest overall scores. However, we observe that LEAST may increase

the score variance of the algorithms (e.g., wider spread and outliers in the box plots), suggesting a potential direction for future work, refining the adaptive evaluation mechanism to enhance training stability.

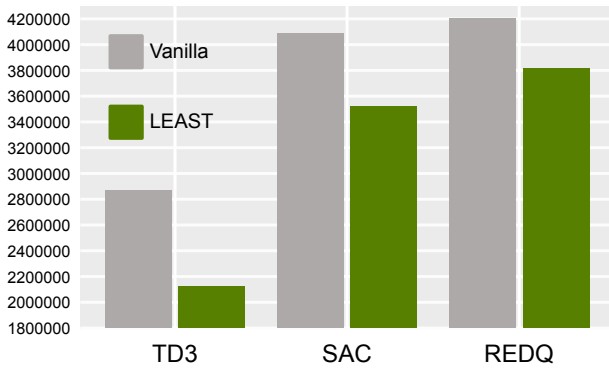

Figure 7. **Comparison of sample efficiency across MuJoCo tasks.** For each algorithm, we report the average number of training steps required to reach the maximum normalized score achieved by its corresponding vanilla (baseline) variant. Lower values indicate faster learning.

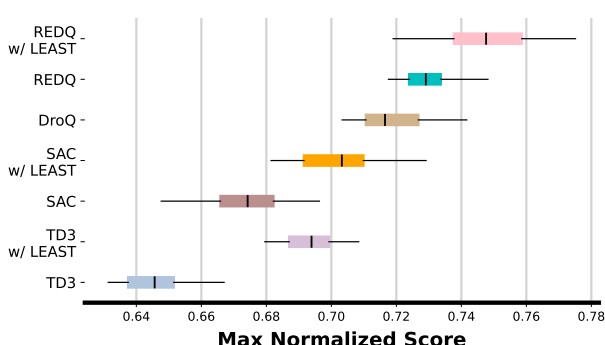

Figure 8. **Final performance on MuJoCo task,** aggregated using the same evaluation metrics as in Figure 4. Results are normalized and averaged over 5 random seeds. Higher values indicate better overall task performance.

### 4.2. Effectiveness on Visual Reinforcement Learning

**Experimental Setup** To evaluate the generalization of LEAST to the visual domain, we adopt DrQv2 (Yarats et al., 2021), a state-of-the-art Vision RL algorithm, as our backbone. We select four image-based control tasks from the DeepMind Control Suite (Tassa et al., 2018) to examine whether the benefits of LEAST transfer to visual RL settings. Additionally, we include several DrQv2-based baselines that introduce observation encoders: CURL (Laskin et al., 2020), A-LIX (Cetin et al., 2022), and TACO (Zheng et al., 2024). Among these, TACO represents a recently proposed variant of DrQv2, which we use as the SOTA

reference. Hyperparameters for DrQv2 are kept consistent across all experiments; see Appendix B for details.

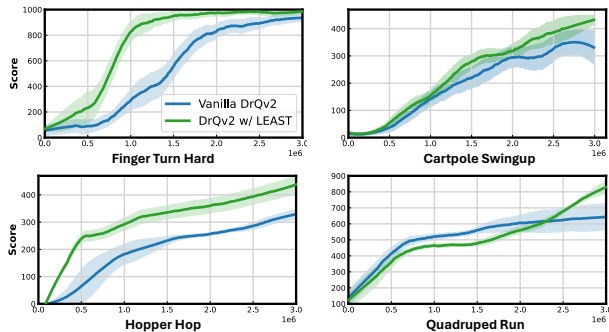

Figure 9. **Performance on image-based control tasks.** The horizontal axis represents training steps; the vertical axis shows the cumulative score. Each curve depicts the mean performance over 5 random seeds, with shaded regions indicating standard deviation

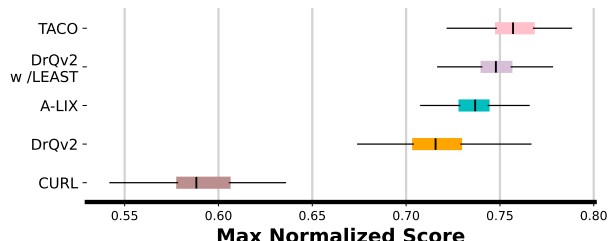

Figure 10. **Final performance on DMC tasks.** Results are aggregated using the same metric as in Figure 4. LEAST performs competitively with feature-based baselines.

Figure 9 shows that although DrQv2 enhances SAC with data augmentation and representation learning, LEAST still provides stable improvements in learning efficiency. For example, in the Finger Turn Hard task, convergence is approximately 30% faster than vanilla DrQv2. Figure 10 reports normalized scores across all tasks. While CURL and A-LIX improve feature representations to accelerate learning, LEAST outperforms both and approaches the performance of TACO. These results suggest that in visual RL, unlocking adaptive stopping can be a promising and cost-effective alternative to heavy architectural modifications, requiring no additional networks to improve sample efficiency.

### 4.3. Robustness of Median Threshold Estimation

As analyzed in §3.2, we evaluate the individual contributions of LEAST components. Here, we compare the use of the median vs. the arithmetic mean to estimate the central tendency of $\mathcal{B}_Q$ on the Ant task. Figure 11 shows that the median yields a more stable estimate, avoiding spikes. This robustness arises because the median is less sensitive to outliers from recent trajectories, thereby mitigating policy collapse caused by frequent early stopping.

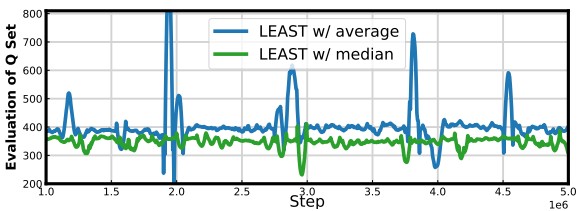

*Figure 11.* **Comparison of methods for estimating the central tendency of $\mathcal{B}_Q$ on the Ant task.** Using the median provides more stable and robust estimates than the arithmetic mean, particularly in the presence of outliers. This stability helps mitigate overestimation in early stopping decisions and improves the reliability of the adaptive threshold used in LEAST.

### 4.4. Sensitivity to the Learning Potential Weight ($\omega$)

We now examine the role of the learning potential weight $\omega$ in LEAST, which modulates the $Q$-value threshold to balance trajectory quality with potential learning value. Figure 12 illustrates the effect of varying the importance weight $\omega$ in the threshold calculation. Compared to TD3, SAC is more sensitive to this parameter. A value in the range $[0.3, 0.6]$ provides a robust trade-off between $\epsilon$ and $\omega$ for both algorithms. This highlights the importance of balancing quality-based stopping with learning potential signals when designing adaptive termination criteria.

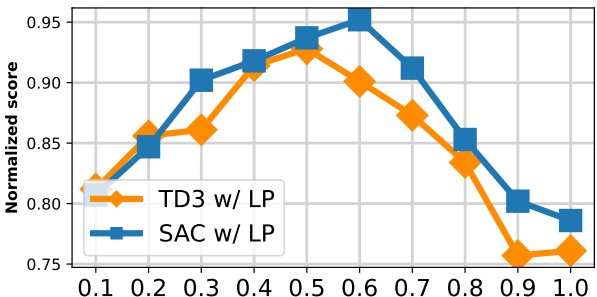

*Figure 12.* **Sensitivity analysis of the weighting factor $\omega$ in the adaptive stopping threshold of LEAST.** We vary the scaling coefficient applied to $\omega$ across a range of values and evaluate its impact on final performance. Results are averaged over 5 seeds. This analysis reveals that LEAST is robust to moderate changes in $\omega$, with the best performance typically achieved when $\omega$ lies in the range $[0.3, 0.6]$.

### 4.5. Effect of Adaptive Noise Scaling

This section analyzes the sensitivity of our method to parameters in the noise schedule. Table 1 shows that the optimal value of $\bar{\sigma}$ lies within $[0.25, 0.35]$ for TD3 and $[0.15, 0.25]$ for SAC. We hypothesize that SAC, being a stochastic policy, inherently exhibits behavioral diversity and thus benefits less from large noise. Table 2 evaluates the length threshold $e$. For MuJoCo, a range of $[400, 500]$ is empirically robust with $m = 50$ episodes used for statistical estimation.

| Task | 0.15 | 0.25 | 0.35 | 0.45 |
|------|------|------|------|------|
| TD3 | $0.684 \pm 0.026$ | $0.695 \pm 0.035$ | $0.694 \pm 0.028$ | $0.671 \pm 0.049$ |
| SAC | $0.697 \pm 0.032$ | $0.706 \pm 0.039$ | $0.688 \pm 0.057$ | $0.663 \pm 0.074$ |

*Table 1.* Ablation study of $\bar{\sigma}$. Values average over 5 random seeds.

| Task | 300 | 400 | 500 | 600 |
|------|-----|-----|-----|-----|
| TD3 | $0.681 \pm 0.062$ | $0.685 \pm 0.028$ | $0.696 \pm 0.037$ | $0.679 \pm 0.053$ |
| SAC | $0.692 \pm 0.038$ | $0.706 \pm 0.039$ | $0.696 \pm 0.034$ | $0.701 \pm 0.036$ |

*Table 2.* Ablation study of $e$. Values average over 5 random seeds.

### 4.6. Impact of Entropy-Guided Buffer Resizing

This section evaluates the sensitivity of LEAST to the entropy-based dynamic scaling parameter $\gamma$. We test this using TD3 and DrQv2 across all scenarios. Table 3 shows that $\gamma \in [0, 0.1]$ yields robust performance. We attribute this to the instability of early-stage trajectories, where larger $\gamma$ may mischaracterize the behavioral distribution due to noise.

| Task | 0 | 0.05 | 0.10 | 0.15 | 0.20 |
|------|---|------|------|------|------|
| DMC | $0.724 \pm 0.047$ | $0.743 \pm 0.052$ | $0.756 \pm 0.076$ | $0.712 \pm 0.056$ | $0.715 \pm 0.042$ |
| MuJoCo | $0.683 \pm 0.043$ | $0.706 \pm 0.039$ | $0.703 \pm 0.041$ | $0.702 \pm 0.021$ | $0.688 \pm 0.064$ |

*Table 3.* Ablation study of $\gamma$. Values average over 5 random seeds.

### 4.7. Influence of Start Time and Initial Set Size

Table 4 analyzes the impact of the LEAST start time. For MuJoCo tasks, initiating at 10–20% of training is optimal, while visual tasks (e.g., DMC) benefit from earlier starts (5–15%). Table 5 investigates the effect of the initial set size for $\mathcal{B}_Q$. We recommend using 250 episodes. Too small a set may lead to poor statistical estimates, while too large may over-smooth recent behavior.

| Task | 0.25M | 0.5M | 0.75M | 1M |
|------|-------|------|-------|-----|
| Ant | $5492.48 \pm 318.71$ | $6647.57 \pm 441.39$ | $6836.36 \pm 305.43$ | $6408.22 \pm 246.13$ |
| Quadruped run | $947.61 \pm 32.25$ | $953.28 \pm 26.83$ | $937.44 \pm 54.75$ | $916.52 \pm 44.16$ |

*Table 4.* Ablation study of start time. Values average over 5 random seeds.

| Task | 50 | 150 | 250 | 350 |
|------|-----|-----|-----|-----|
| Ant | $4718.47 \pm 1069.37$ | $6310.15 \pm 726.05$ | $6836.36 \pm 305.43$ | $6792.46 \pm 327.58$ |
| Quadruped run | $751.33 \pm 86.53$ | $941.37 \pm 25.35$ | $953.28 \pm 26.83$ | $951.07 \pm 30.49$ |

*Table 5.* Ablation study of initial set size. Values average over 5 random seeds.

## 5. Related Work

**Sample Efficiency in Deep RL** Clipped Double Q-learning (Fujimoto et al., 2018) is a landmark approach that introduces a double $Q$ network to mitigate overestimation in reinforcement learning (Hasselt, 2010; Van Hasselt

et al., 2016; Song et al., 2019; Pan et al., 2019; 2020; 2021). Building on this, Moskovitz et al. (2021) proposed an online method to tune pessimism and correct for the underestimation introduced by TD3. Nauman et al. (2024) demonstrated that using a residual network backbone can further alleviate pessimistic $Q$-learning. Chen et al. (2021) addressed estimation bias by increasing the number of $Q$ networks to an ensemble of ten. Other studies have shown that scaling up DQN architectures (Schwarzer et al., 2023; Obando-Ceron et al., 2024b;a; Sokar et al., 2025) and increasing the replay ratio (D'Oro et al., 2022) can also accelerate $Q$-function learning.

Another line of work focuses on optimistic exploration strategies (Wang et al., 2019; Moskovitz et al., 2021). For instance, ICM (Pathak et al., 2017) uses forward prediction error as a curiosity signal to encourage exploration of novel states. Recent research also advocates the use of generative models such as VAEs (Imre, 2021), GANs (Huang et al., 2017; Daniels et al., 2022), and diffusion models (Lu et al., 2024; Yu et al., 2023) to augment data and meet sampling requirements. However, existing methods largely overlook the impact of sunk cost fallacy in deep RL. This paper aims to fill that gap by proposing adaptive stop sampling as a cost-effective strategy to improve the performance of deep RL algorithms, a direction that has been largely underexplored by the community.

**Early Stopping in Deep RL** To the best of our knowledge, there is limited work on adaptive termination in online deep RL. Existing studies primarily examine performance in environments with uncertain episode lengths or time limits (Pardo et al., 2018). Poiani et al. (2023; 2024) theoretically demonstrate that truncating trajectories based on prior experience can accelerate learning in Monte Carlo methods. Mandal et al. (2023) show that policies can benefit from episode lengths that increase linearly in extended-duration tasks. In the context of Constrained Markov Decision Processes (CMDPs), some works have explored early termination based on prior safety signals or contextual knowledge (Sun et al., 2021), as well as using offline data to estimate reward distributions (Killian et al., 2023), with applications in domains such as healthcare (Fatemi et al., 2021). These insights on how trajectory length influences policy learning provide useful foundations and motivation for our research.

## 6. Future Work and Limitations

We encourage the community to further investigate the sunk cost bias in reinforcement learning. Based on our findings, we identify three promising directions for future work: (i) While LEAST improves performance, stability can still be enhanced. Future research could focus on developing more sensitive and robust evaluators to improve stopping decisions in complex scenarios, e.g., optimizing the combination of $Q$-value and loss to more accurately characterize trajectory quality. (ii) As shown in Figure 3, the proportion of samples with high loss but low $Q$-value (blue region) remains largely unchanged. These are likely low-quality transitions with minimal expected return. Future work could investigate refining the sampling threshold by incorporating additional indicators to better filter out such data. (iii) After reset, agents may re-enter previously suboptimal trajectories. Introducing a noise-based diversification module could help, though care must be taken to properly schedule the noise magnitude and timing to balance exploration with training stability.

To ensure consistency and reduce computational demands, we adopted the default hyperparameters provided by each baseline model for all experiments. While this choice facilitates reproducibility, it is well known that RL algorithms can exhibit significant sensitivity to hyperparameter settings (Ceron et al., 2024; Ceron & Castro, 2021). A thorough hyperparameter sweep for each experimental configuration would be preferable in principle, but remains impractical given the associated computational expense

## 7. Conclusion

In this paper, we highlight the presence of sunk cost fallacy in mainstream deep RL algorithms. This fallacy leads agents to persist in executing entire episodes, even when trajectories become suboptimal, resulting in unnecessary interactions and buffer contamination, ultimately hindering learning. We identify this as a previously overlooked limitation in the literature: the inability of current algorithms to terminate bad trajectories early may significantly impact overall performance. To address this, we propose LEAST, a direct optimization approach that quantifies trajectory quality and enables early stopping, without requiring additional network components. LEAST effectively mitigates the sunk cost fallacy and improves learning efficiency across a variety of tasks. We hope that our findings will inspire further research aimed at optimizing sampling and learning efficiency.

## Acknowledgment

This work is supported by the National Natural Science Foundation of China 62406266. The authors would like to thank Roger Creus and Gopeshh Subbaraj for valuable discussions during the preparation of this work. We thank the anonymous reviewers for their valuable help in improving our manuscript. We would also like to thank the Python community (Van Rossum & Drake Jr, 1995; Oliphant, 2007) for developing tools that enabled this work, including NumPy (Harris et al., 2020) and Matplotlib (Hunter, 2007).

## Impact Statement

This paper presents work whose goal is to advance the field of Machine Learning. There are many potential societal consequences of our work, none of which we feel must be specifically highlighted here.

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

# A. Related Preliminaries

## A.1. Baselines

**TD3** In our paper, we utilize TD3 as the representative of a deterministic policies. TD3, an Actor-Critic algorithm, is widely used in various decision scenarios as the baseline (Li et al., 2021; Liu et al.), and has derived a wide range of variants to establish new SOTA many times. Differs from the traditional policy gradient method, DDPG (Lillicrap et al., 2015), TD3 employs two heterogeneous critic networks, denoted as $Q_{\theta_{1,2}}$ to mitigate the issue of over-optimization in Q-learning. Thus, the Loss function of critics is:

$$L_Q(\theta_i) = \mathbb{E}_{s,a,s'}\big[(y - Q_{\theta_i}(s,a))^2\big] \text{ for } \forall i \in \{1,2\}. \tag{5}$$

Where $y = r + \gamma \min_{j=1,2} Q_{\bar{\theta}_j}(s', \pi_{\bar{\phi}}(s'))$, $\bar{\phi}$ denotes the target network parameters. The actor is updated according to the Deterministic Policy Gradient (Fujimoto et al., 2018):

$$\nabla_\omega J(\phi) = \mathbb{E}_s\big[\nabla_{\pi_\phi(s)} Q_{\theta_1}(s, \pi_\phi(s)) \nabla_\phi \pi_\phi(s)\big]. \tag{6}$$

**SAC** We choose SAC (Haarnoja et al., 2018) as a representative of the stochastic policy in combination with `LEAST` in the main experiment. SAC is designed to maximize expected cumulative rewards while also promoting exploration through the principle of maximum entropy. The actor aims to learn a stochastic policy that outputs a distribution over actions, while the critics estimate the value of taking a particular action in a given state. This allows for a more diverse set of actions, facilitating better exploration of the action space. In traditional reinforcement learning, the objective is to maximize the expected return. However, SAC introduces an additional term that maximizes the entropy of the policy, encouraging exploration. The goal is to optimize the following objective:

$$J(\pi) = \mathbb{E}_{s_t, a_t}[r(s_t, a_t) + \alpha H(\pi(\cdot|s_t))]$$

$H(\pi(\cdot|s_t)$ is the entropy of the policy. $\alpha$ is a temperature parameter that balances the trade-off between the reward and the entropy. The training process for SAC consists of two main updates: updating the value function and updating the policy. The value function is updated by minimizing the following loss:

$$L(Q) = \mathbb{E}_{(s,a,r,s')\sim D}\big[\frac{1}{2}(Q(s,a) - (r + \gamma V(s')))^2)\big]$$

$\gamma$ is the discount factor, dictating how much future rewards are valued. $V(s')$ is the value function of the next state, typically approximated using a separate neural network. The policy is updated by maximizing the following objective:

$$J(\pi) = \mathbb{E}_{s_t\sim D}[\mathbb{E}_{a_t\sim\pi}[Q(s_t, a_t) - \alpha \log \pi(a_t|s_t)]]$$

$-\alpha \log \pi(a_t|s_t)$ represents the entropy of the policy, promoting exploration.

**REDQ** Chen et al. (2021) found that the accumulation of the learned Q functions' estimation bias over multiple update steps in SAC can destabilize learning. To remedy this bias, they increase the number of Q networks from 2 to an ensemble of 10. Its training procedure is consistent with SAC. REDQ is recognized as an effective algorithm to improve the sampling efficiency of deep RL.

**DrQv2** DrQv2 is a model-free off-policy algorithm for image-based continuous control. DrQ-v2, an actor-critic approach that uses data augmentation to learn directly from pixels. We introduce several improvements including: switch the base RL learner from SAC to DDPG, Incorporating n-step returns to estimate TD error. They also find it useful to apply bilinear interpolation on top of the shifted image for the random image augmentation and to optimize the exploration schedule.

**A-LIX & TACO** In this paper we chose two well-known variants of DrQv2 as the baseline to evaluate the research value of adaptive stopping as a new research line. A-LIX try to mitigate catastrophic self-overfitting in DrQv2 from the feature representation level, that is, providing adaptive regularization to the encoder's gradients to stabilize temporal difference learning from encoders. And it improves the performance of DrQv2 on diverse image input tasks.

TACO also optimize DrQv2 by adding an additional feature extraction model. Practivally, TACO incorporates an auxiliary temporal action-driven contrastive learning objective to learn state and action representations through mutual information, and self-prediction representations. It established a new SOTA on DMC and metaworld benchmarks.

### A.2. Maze

We use the widely used open-source Point Maze scenario to build the three tasks of small, medium and large. The specific layout is shown in Figure 13.

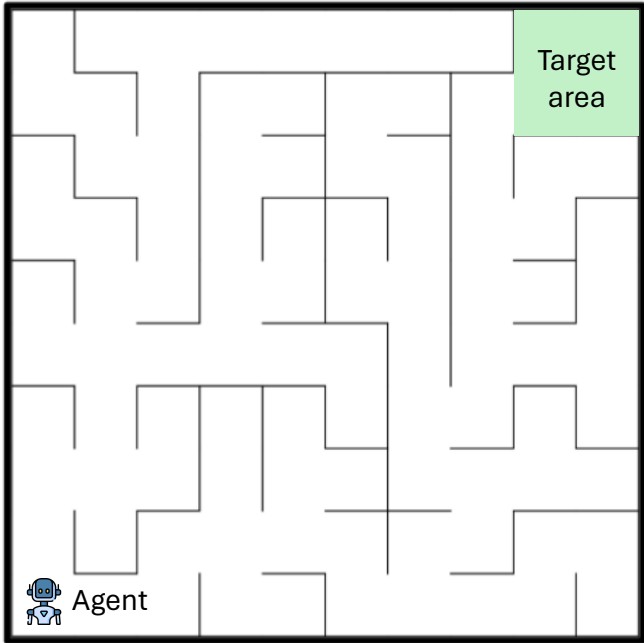

*Figure 13.* Layout of the maze.

The agent, a 2D point, is noisily initialized in the bottom left. At test time, we evaluate it on reaching the right corner. The 2-D state and action space correspond to the planar position and velocity, respectively. The episode length is $50$ steps. The top right corner is the hardest to reach, so we set this to be the test-time goal. We increase the point size to $1.25\times$, with a speed factor of $0.9$. The dense reward is designed according to the L2 distance from the center of the target region to the point. The parameters for SAC are consistent with the recommendations, with $\gamma$ set to $0.95$.

We further count the final positions of the last 50 episodes of each run for the vanilla agent as well as the advanced agent (`LEAST`) in Figure 14. The results show that the advanced agent can effectively avoid multiple dead ends and successfully reach the target area under the limited interaction budget.

### A.3. Measurement

**Fraction of Active Units (FAU)** Although the complete mechanisms underlying plasticity loss remain unclear, a reduction in the number of active units within the network has been identified as a key factor contributing to this deterioration in both visual RL (Ma et al., 2023; Sokar et al., 2023) and traditional RL (Abbas et al., 2023; Liu et al., 2025). Consequently, the fraction of active units (FAU) is widely utilized as a metric for assessing plasticity. Specifically, the FAU for neurons located in layer $\mathcal{N}$, denoted as $\psi^{\mathcal{N}}$, is formally defined as:

$$\psi^{\mathcal{N}} = \frac{\sum_{n \in \mathcal{N}} 1(a_n(x) > 0)}{N}$$

where $a_n(x)$ represent the activation of neuron n given the input $x$, and $n$ is the neuron within layer $\mathcal{N}$. The meaning of this metric is the proportion of the number of activated parameters to the total number of parameters. That is to say, the slower this ratio decreases, the slower the plasticity loss of the network is, so as to maintain the representation ability of the quantity network for new data. This allows it to be visually visualized following the training time steps, which can be used to evaluate the continuous learning ability of deep RL agents.

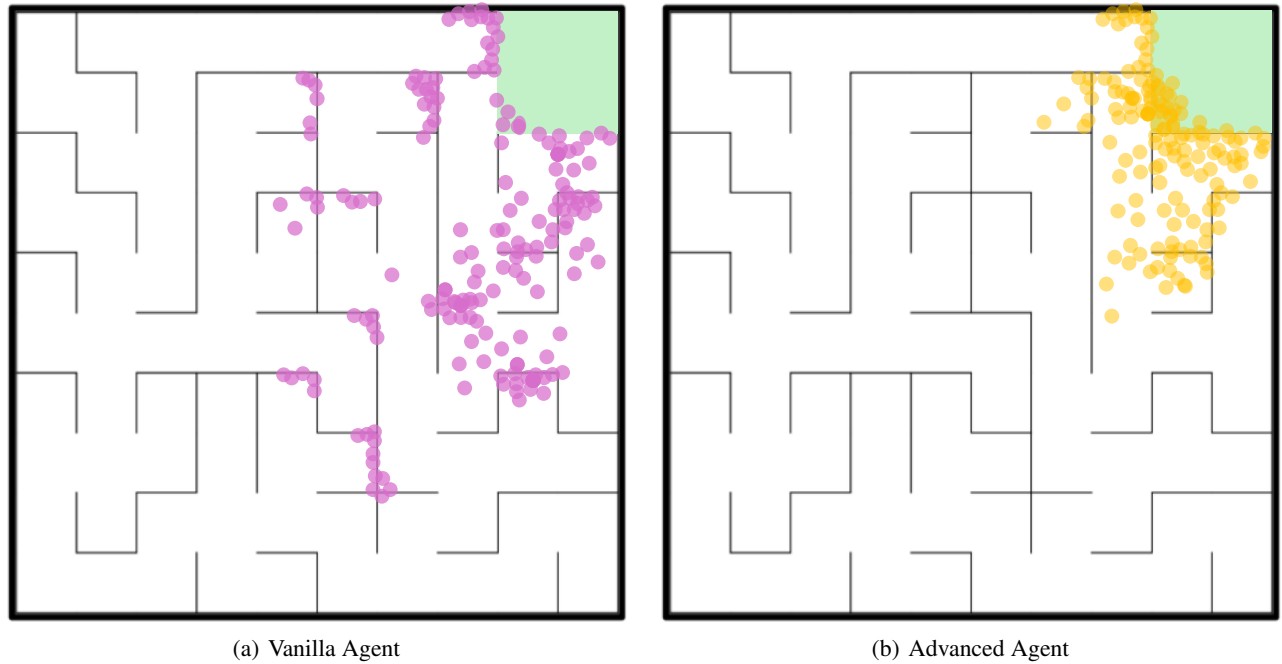

(a) Vanilla Agent  (b) Advanced Agent

*Figure 14.* Statistics of the final positions of two agents on the medium task.

### A.4. Sunk Cost Fallacy

Sunk cost fallacy (Turpin et al., 2019), also known as sunk cost bias (Arkes & Blumer, 1985), is widely discussed in cognitive science. It describes the tendency to follow through with something that people have already invested heavily in (be it time, money, effort, or emotional energy), even when giving up is clearly a better idea. This phenomenon is particularly prevalent in the world of investments (Dijkstra & Hong, 2019). Whether it's in the stock market, real estate, or even a business venture, investors often fall into the trap of holding onto underperforming assets because they've already sunk so much money into them (Haita-Falah, 2017). In this paper, we investigate whether deep RL agents deployed on cumulative reward maximization tasks also suffer from sunk cost fallacy.

## B. Experimental Details

### B.1. Structure

**TD3** In this paper, we use the official architecture, and the detailed structure is shown in Tab 6.

| Layer | Actor Network | Critic Network |
|---|---|---|
| Fully Connected | (state dim, 256) | (state dim, 256) |
| Activation | ReLU | ReLU |
| Fully Connected | (256, 128) | (256, 128) |
| Activation | ReLU | ReLU |
| Fully Connected | (128, action dim) and (128, 1) | |
| Activation | Tanh | None |

*Table 6.* Network Structures for TD3

**SAC & REDQ** In this paper, we follow the setting that commonly used in Dep RL community, and the detailed structure is shown in Tab 7. We set the number of ensembles to 10 according to the official setting of REDQ https://github.com/BY571/Randomized-Ensembled-Double-Q-learning-REDQ-/blob/main, but for fairness, the UTD of all algorithms is set to 1.

| Layer | Actor Network | Critic Network |
|---|---|---|
| Fully Connected | (state dim, 256) | (state dim, 256) |
| Activation | ReLU | ReLU |
| Fully Connected | (256, 128) | (256, 128) |
| Activation | ReLU | ReLU |
| Fully Connected | (128, $2\times$ action dim) and (128, 1) | |
| Activation | Tanh | None |

*Table 7.* Network Structures for SAC

**DrQv2& TACO& A-LIX** We use these three methods to conduct experiments on robot control tasks within DeepMind Control using image input as the observation. All experiments are based on official codes. Code of DrQv2 can be found at `https://github.com/facebookresearch/drqv2`, the website of TACO is `https://github.com/FrankZheng2022/TACO` and A-LIX code can be downloaded at `https://github.com/Aladoro/Stabilizing-Off-Policy-RL/tree/master`. We maintain the unchanged architecture from the official setting.

## B.2. Implementation Details

Our code is implemented with Python 3.8 and Torch 1.12.1. All experiments were run on NVIDIA GeForce GTX 3090 GPUs. Each single training trial ranges from 10 hours to 21 hours, depending on the algorithms and environments, e.g. DrQv2 spends more time than TD3 to handle the image input and DMC needs more time than OpenAI mujoco for image rendering.

**TD3** Our TD3 is implemented with reference to `github.com/sfujim/TD3` (TD3 source-code). The hyper-parameters for TD3 are presented in Table 8. Notably, for all OpenAI MuJoCo experiments, we use the raw state and reward from the environment, and no normalization or scaling is used. An exploration noise sampled from $N(0, 0.1)$ (Lillicrap, 2015) is added to all baseline methods when selecting an action. The discounted factor is 0.99 and we use Adam Optimizer (**?**) for all algorithms. Tab.8 shows the hyperparameters of TD3 used in all our experiments. We suggest using seed $0 \rightarrow 7$ to reproduce the learning curve in the main text.

| Hyperparameter | TD3 | TD3 w/ LEAST |
|---|---|---|
| Actor Learning Rate | $1e^{-4}$ | $1e^{-4}$ |
| Critic Learning Rate | $1e^{-3}$ | $1e^{-3}$ |
| Discount Factor | 0.99 | 0.99 |
| Batch Size | 128 | 128 |
| Buffer Size | $1e6$ | $1e6$ |

*Table 8.* Shared hyperparameter setting of TD3.

**SAC** Our TD3 is implemented based on `https://github.com/tyq1024/RLx2/tree/master/RLx2_SAC`, a reliable open source implement in Pytorch style). The hyper-parameters for SAC are recorded in Table 9.

**DrQv2** We use DrQv2 as the backbone to verify our methods on DeepMind Control tasks. The details can be seen in Tab.10. We follow the setting used in DrQv2, for Walker Stand/Walk/Run tasks, we set the mini-batch size as 512 and n-step return of 1. In the *Quadruped Run* task, we use a replay buffer of size $10^5$ and set the learning rate to $8 \times 10^{-5}$. We also increase the feature dimensionality to 100 in *Humanoid* scenarios.

**Lern to Stop** For our Learn to stop schedule in TD3, we generally set the start time as $0.6M$ step for most OpenAI MuJoCo tasks, expect Ant (start at $0.75M$)and HalfCheetah (start at $0.25M$). For our Learn to stop schedule in SAC (REDQ), we set the start time as $0.5M$ step for most OpenAI MuJoCo tasks, expect Ant (start at $0.65M$)and HalfCheetah (start at $0.25M$). And for DMC tasks, the schedule will be turn on at $0.4M$ step. As for the initial size of the reflection sets, we set 150 for both TD3 and SAC for OpenAI muJoCo tasks, and 250 for image input tasks. The scale of Loss ratio $\omega$ is set as 0.5 for all tasks, but 0.7 for Humanoid and 0.6 for HalfCheetah in SAC. The max threshold is the max Q value in $\mathcal{B}_Q$

| Hyperparameter | SAC | SAC w/ LEAST |
|---|---|---|
| Optimizer | Adam | Adam |
| Learning rate | $3 \times 10^{-4}$ | $3 \times 10^{-4}$ |
| Discount factor | 0.99 | 0.99 |
| Number of hidden layers | 2 | 2 |
| Number of hidden units per layer | 256 | 256 |
| Activation Function | ReLU | ReLU |
| Batch size | 256 | 256 |
| Warmup steps | 5000 | 5000 |
| Target update rate | 0.005 | 0.005 |
| Target update interval | 1 | 1 |
| Actor update interval | 1 | 1 |
| Entropy target | $-|A|$ | $-|A|$ |

*Table 9.* Shared hyperparameter setting of SAC.

| Hyperparameter | DrQ | DrQ w/ LEAST |
|---|---|---|
| Replay buffer capacity | $1e6$ | $1e6$ |
| Action repeat | 2 | 2 |
| Seed frames | 4000 | 4000 |
| Exploration steps | 2000 | 2000 |
| n-step returns | 3 | 3 |
| Mini-batch size | 256 | 256 |
| Discount $\gamma$ | 0.99 | 0.99 |
| Optimizer | Adam | Adam |
| Learning rate | $1e-4$ | $1e-4$ |
| Agent update frequency | 2 | 2 |
| Critic Q-function soft-update rate | 0.01 | 0.01 |
| Features dim. | 50 | 50 |
| Hidden dim. | 1024 | 1024 |
| Exploration stddev. clip | 0.3 | 0.3 |
| Exploration stddev. schedule | $linear(1.0, 0.1, 500000)$, Humanoid tsks: $linear(1.0, 0.1, 2000000)$ | $linear(1.0, 0.1, 500000)$, Humanoid tsks: $linear(1.0, 0.1, 2000000)$ |

*Table 10.* A default set of hyper-parameters used in our DrQv2-based methods.

(we find that $0.75*$median $Q$ can aslo be a magic number) and the min threshold is set as min value in $\mathcal{B}_Q$ ($0.25*$ median $Q$ is a also good choice). Too many sensitive parameters are also an issue to be optimized in the future of this work. To prevent extreme threshold adjustments, we use $Clip$ function to limit $\omega_i \times \epsilon \in [Q_{min}[i], Q_{max}[i]]$, where $Q_{max}[i]$ denotes the highest $Q$ value among intra-episode step $i$, $Q_{min}[i]$ is the min value.

## C. Additional Experiments

### C.1. Effects on Continuous Learning and Plasticity

Nikishin et al. (2022) finds that the rapid loss of plasticity in network caused by overfitting to noisy data in the early training is a key reason for losing continuous learning ability. In §3.2, we observe that LEAST can effectively optimize the data distribution in the buffer, thus avoiding the agent from training via useless (noisy) transitions too often. This may go some way to overcoming aformentioned issue and maintaining the continuous learning ability (mitigating the loss of network plasticity).

In this section, we verify in detail whether the above advantage exists in our method. We conduct a evaluation on the difficult long-time learning task – humanoid walk, i.e. DrQv2 needs about $6\times$ steps than normal DMC tasks to learn a sequential of skills ($15M$). The parameters remain the same as in §4.2. Notably, we introduce a widely used metric,the Fraction of Active Units (**FAU**[4]), to quantify the plasticity of the network. FAU is used in such a way that the gentler the downward trend of the curve indicates the better-maintained plasticity of the network. Figure 15 shows that our method decreases FAU slower than the baseline and achieves impressive performance on long-term learning tasks. This corroborates the positive effect of LEAST on maintaining the continuous learning ability. In the future, we will deeply analyze the theoretical correlation between plasticity and adaptive stopping.

---

[4]Plasticity measurement is introduced in Appendix A.3.

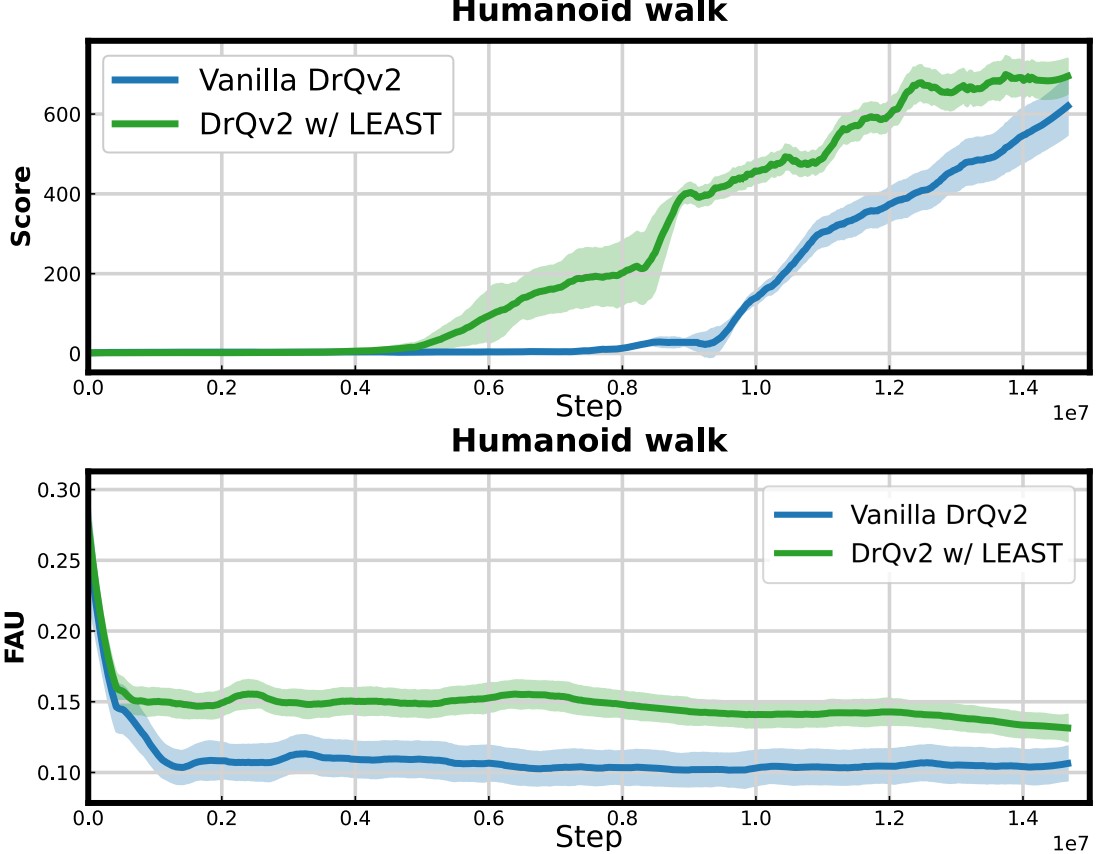

*Figure 15.* Performance on long-term training task. The curve and shade denote the mean and the standard deviation over 5 runs.

### C.2. Ablation Study of Update-to-Data

We use TD3 as the backbone and try to improve UTD to enable fast iteration of policies to better capture the advantages of LEAST. However, it can be seen from Figure 16 that with the increase of UTD, the efficiency of policy optimization becomes slower, which we think is due to the inaccuracy of the fitting of $Q$ function by basic algorithms such as TD3, which cannot adapt to high UTD (Chen et al., 2021).

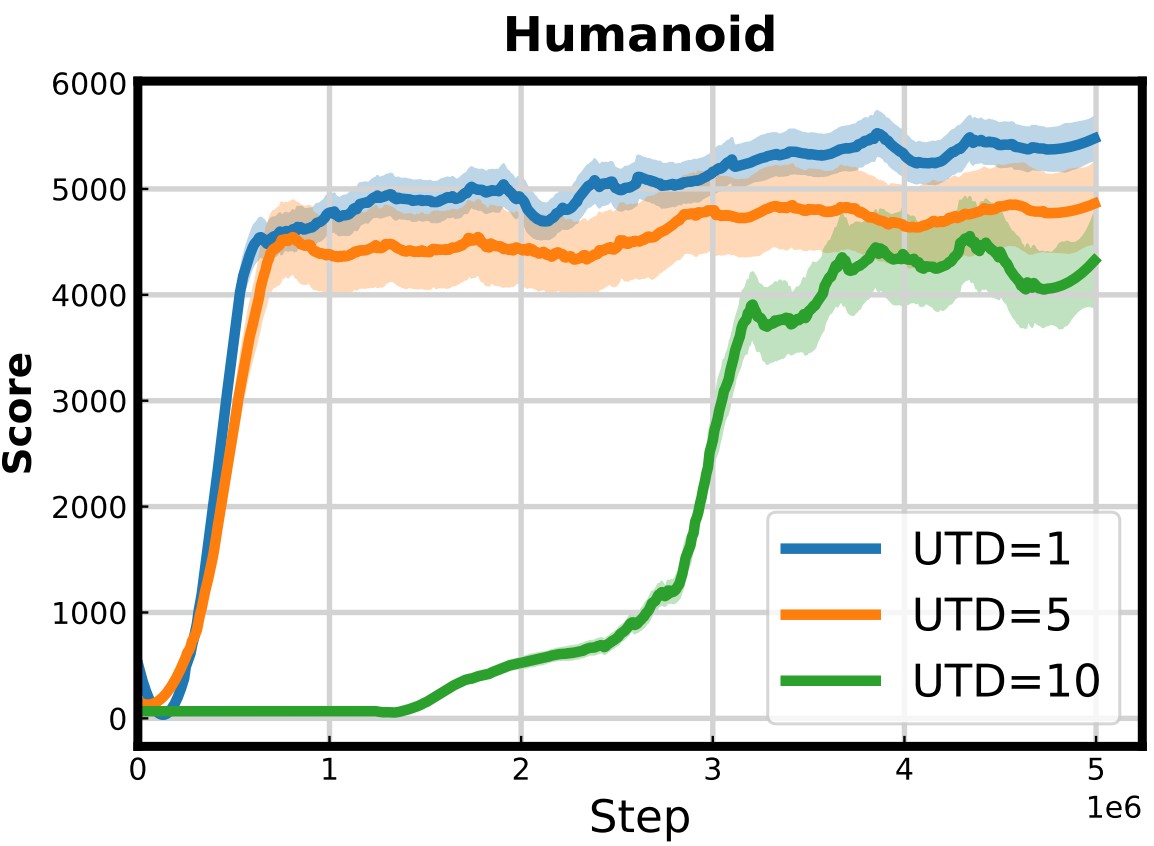

*Figure 16.* Sensitivity to UTD. Average of 5 runs on Ant.

