# OpenReview forum: "The Courage to Stop: Overcoming Sunk Cost Fallacy in Deep Reinforcement Learning"
_ICML.cc/2025/Conference — ICML 2025 poster_

### Official Review · Reviewer_4RG4 · 2025-03-04

**Overall Recommendation:** 4

**Summary:**

The paper proposes a simple technique to avoid the issue of deep RL agents collecting familiar low-rewarding data to avoid the issue of re-sampling this data during training as a way to improve sample efficiency of deep RL agents. The paper integrates the algorithm in a variety of algorithms, applies it on different benchmarks including image data, and conducts an ablation.

## update after rebuttal

I am leaning on acceptance since my questions were adequately addressed and the paper presents an interesting idea.

**Claims And Evidence:**

Yes, in general, paper's claims are accompanied with convincing evidence.
Few issues:
- The graph on Figure 7 does not include any information about standard deviation, variance etc. Nor does it mention the number of trials, which I am assuming 5 * # of environments. This information is critical even though LEAST clearly has better sample efficiency.
- The discussion on entropy aware dynamic buffer size and adjusting exploration noise (Section 3) seems quite ad-hoc. The previous paragraphs are well-motivated and those are well-analyzed in the empirical section, but these other sections seem out-of-place. I see there is some evaluation in the Appendix. I think the main paper should have some discussion about this.
- Section 4.3 mentions that LEAST helps “avoid the agent from training via useless transitions too often”. While LEAST does help plasticity it is a little unclear to me whether this is because it fits higher quality data or because there are fewer data samples since LEAST filters them out. It seems like an algorithm’s plasticity could increase if it just updated less often. I presume the number of gradient steps are the same, which means it would be the former. If this is the case, this should be specified.

**Essential References Not Discussed:**

It would be useful to make some references to algorithms that sample data differently instead of filtering data. For example, prioritized experience replay (Schaul 2015) or event tables (Kompella 2023). These methods seems similar in spirit of updating the agent using better data.

**Experimental Designs Or Analyses:**

Yes, the paper conducts a reasonable analysis except for one point mentioned above (copy-pasting here): The graph on Figure 7 does not include any information about standard deviation, variance etc. Nor does it mention the number of trials, which I am assuming 5 * # of environments. This information is critical even though LEAST clearly has better sample efficiency.

**Methods And Evaluation Criteria:**

Yes, the paper integrates the proposed algorithm in different settings, baseline algorithms (TD3, SAC, DrQv2), and in visual/non-visual environments that are commonly used in the deep RL literature.

**Other Comments Or Suggestions:**

Page 6 paragraphs have some typos and the dynamic buffer size/exploration noise paragraphs are unclear.

**Other Strengths And Weaknesses:**

Strengths:
- The algorithm is simple to implement and incorporate into any RL algorithm
- It is evaluated on visual and non-visual environments with different baselines.
- It conducts a thorough empirical analysis and ablation

Weaknesses:
- See above
- The last two lines in the first paragraph Section 3.2 are very unclear and seem to just be filled with jargon. It would be better to simplify.
- The paragraphs on dynamic buffer size and exploration noise are quite confusing. The notation used also is slightly abused which makes things tricky to parse. Are the last h episodes removed? If not, which ones? What does UTD mean? It should be specified.
- First few lines on page 6 discusses why LEAST eventually does better in HalfCheetah, but it does not discuss why it may be worse during early training. I think that is the real anomaly which should be explained.

**Questions For Authors:**

- Given that the training loss is so critical to filter data, what do the authors think about the unreliability of the loss function in off-policy algorithms and how that may be affecting LEAST [1]?
- How do you think performance would change if instead of storing the Q values in B_Q, the Monte Carlo returns were saved?
- Isn’t the issue of unchanging policy true for even non-LEAST methods? It's a little unclear to me why LEAST has this issue. Given this seems like a universal issue, it seems like LEAST may be better since it has an improved exploration strategy.
- Nothing to really comment but something of potential interest to authors is [2], which is for the average reward setting where the agent tries to figure out when it should reset the episode, where there is a cost associated with resetting the episode.


[1] Why Should I Trust You, Bellman? The Bellman Error is a Poor Replacement for Value Error. Fujimoto et al. 2022.

[2] RVI-SAC: Average Reward Off-Policy Deep Reinforcement Learning. Hisaki et al. 2024.

**Relation To Broader Scientific Literature:**

The paper falls under the broad category of improving the sample efficiency of RL algorithms in robotics simulation tasks. It revisits how deep RL algorithms are fundamentally designed to use the data they collect and suggests a data filtration process to use higher quality data during training.

**Theoretical Claims:**

N/A

---

> ### Author Rebuttal · Authors · 2025-04-01
>
> Thanks for the insightful suggestions and helpful feedback of our work, and for the recognition of our work's motivation, performance, and potential academic impact.
>
> **Q1:  (1) Given that the training loss is so critical to filter data, what do the authors think about the unreliability of the loss function in off-policy algorithms and (2) how that may be affecting LEAST ?**
>
> (1) To mitigate the unreliability of loss calculations, LEAST mitigates this issue by using the median of the loss values rather than relying on a single loss value per transaction, which reduces sensitivity to noise & biases. It will also be an interesting future direction to consider more robust loss calculation.
>
> (2) While LEAST shows strong performance in the current experimental settings, an interesting direction for future work would be to investigate its behavior in tasks requiring even longer-term learning. In such scenarios, the potential instability of loss calculations might affect the episode-stopping threshold, which could slightly reduce the method’s efficiency in certain edge cases. It is an interesting future direction for further improving the robustness of loss calculations, which can further extend LEAST's applicability and improve its stability across a wider range of tasks.
>
> **Q2: How do you think performance would change if instead of storing the Q values in B_Q, the MC returns were saved?**
>
> Thank you for the suggestion! MC returns could be a potential metric. However, it can introduce two main challenges for implementing this within LEAST’s framework:
>
> - *Interaction efficiency:* MC return requires repeatedly resetting to specific states to compute estimates from multiple rollouts to reduce variance in its estimation, which can be sample-inefficient.
> - *Suffer from biased estimation:* MC return heavily depends on observing complete trajectories up to their true terminal states. When trajectories are truncated prematurely (as in LEAST), the missing components of the return are treated as zero, leading to biased and inaccurate calculations.
>
> We hope future work will explore more advanced estimations based on MC returns to effectively address the aforementioned challenges.
>
> **Q3: Isn’t the issue of unchanging policy true for even non-LEAST methods? It's a little unclear to me why LEAST has this issue. Given this seems like a universal issue...**
>
> While this problem is indeed common in RL, it is particularly critical for LEAST due to its unique mechanism.
>
> The motivation for proposing a noise adjustment schedule stems from specific observations during our experiments. Although the Q- & loss-based stopping threshold provides core benefits, LEAST occasionally triggers frequent resets during early training stages, which can hinder long-term exploration (crucial for extended execution tasks like humanoid walk).
>
> Thus, we introduced a simple yet effective noise mechanism controlled by premature reset frequency, aiming to stabilize exploration, especially during early training. When integrated with TD3, the results showed minimal improvement, suggesting that noise adjustment may be less beneficial for vanilla deep RL compared to its impact on LEAST.
>
> Table. Performance $\uparrow$ (Average of 3 runs):
>
> | Method | Ant | Humanoid |
> | --- | --- | --- |
> | **TD3 w/ noise adjustment** | $5704 \pm 145.23$ | $4591.28\pm 224.82$ |
> | TD3 | $5682.13\pm 129.48$ | $4616.54\pm 273.69$ |
>
> **Q4: Nothing to really comment but something of potential interest to authors ...**
>
> This is very interesting! We will analyze it in the related work of the new version.
>
> **W1: Fig 7 does not include information about std, variance etc... even though LEAST clearly has better sample efficiency.**
>
> Thanks for your suggestion. We used 5 random seeds, and we will update the caption and add error bars in Fig 7.
>
> **W2: The discussion on entropy aware dynamic buffer size & adjusting exploration noise seems quite ad-hoc... should have some discussion**
>
> To ensure robust episode truncation in complex environments, we implement two key mechanisms beyond the core Q-value & loss thresholds: (1) a dynamic buffer that filters outliers to improve threshold stability & accuracy when Q-values become unreliable, and (2) to prevent exploration limitations from frequent resets, we introduce an action noise mechanism that adaptively balances exploration and stopping behavior.
>
> We will elaborate more in the revision.
>
> **W3: … While LEAST does help plasticity it is a little unclear whether this is because it fits higher quality data or .... this should be specified.**
>
> We would like to clarify that LEAST improves plasticity not by reducing the number of updates, instead, it helps the agent to focus on more informative samples which accelerates adaptation and helps it explore better policies.
>
> ---
> We would like to thank the reviewer once again for the time and effort in reviewing our work! We are happy to provide further clarification if you have any additional questions.

---

> > ### Comment · Reviewer_4RG4 · 2025-04-02
> >
> > Thank you for your response! The paper is interesting. I think the insight is useful and I appreciate that the first crack at this issue is simple.

---

> > > ### Author Response · Authors · 2025-04-03
> > >
> > > We would like to sincerely thank the reviewer (Reviewer 4RG4) once again for the valuable feedback, as well as for the time and effort dedicated to reviewing our work.
> > >
> > > We are also grateful to Reviewer zKrJ for the additional questions and constructive suggestions (we have to combine the responses here together due to the rebuttal requirements). We greatly appreciate that you found the paper interesting and the insights useful, and we are thankful for your decision to raise the score to accept.
> > >
> > > We will carefully incorporate all the feedback into the revised manuscript, which has greatly helped us improve the quality, clarity, and overall contribution of our work.

---

### Official Review · Reviewer_Rcz7 · 2025-03-10

**Overall Recommendation:** 3

**Summary:**

The paper introduces the learn to stop (LEAST) heuristic for online off-policy reinforcement learning (RL).
The core idea is the proposition of an adaptive stopping mechanism to prohibit including unhelpful-to-the-learning-task transitions to the replay buffer. The heuristics is then experimentally evaluated on a variety of aspects with four MuJoCo tasks and DMC tasks.

## update after rebuttal
Score increased. See rebuttal comment.

**Claims And Evidence:**

The paper states its main contributions as
(1) identifying a critical limitation of current RL frameworks
(2) introduction of a stopping heuristic with general benefits
(3) experimental validation of the proposed heuristic.

Claim (1) seems to be supported by clear and convincing evidence.
Claim (3) is supported by the experiments presented in the paper.

Claim (2) is not entirely clear to me. See also Question for Authors.

**Essential References Not Discussed:**

Not relevant.

**Experimental Designs Or Analyses:**

The experimental verification looks valid for the presented results.

**Methods And Evaluation Criteria:**

Looks appropriate except open question in Question for Authors.

**Other Comments Or Suggestions:**

To me, the connection between the observed problem and the sunk-cost fallacy seems far-fetched. Sunk-cost fallacy depends on the emotional attachment to whatever resource is already sunk. The continuation of an episode until termination stems rather from a missing mechanism instead of an attached to the current episode.

The referencing of subsection with "§" does not fit ICML standards.

Figure 5 and 13 have surprising format sizes.

Typo: mwthod (327).

**Other Strengths And Weaknesses:**

The presented background (often called preliminaries) has nothing to do with "Deep" RL, it describes RL basics. It seems reasonable though that the addressed problem is especially relevant for Deep RL.


Appendix B.4 is an exact repetition of a paragraph in Section 2.

**Questions For Authors:**

What do you mean by "In this paper, we investigate whether deep RL agents deployed on cumulative reward maximization tasks also suffer from sunk cost fallacy."?

What exactly makes the introduced heuristic special for deep RL in comparison to its importance for "non-deep" RL?

What do you mean by "In this section, we conduct comprehensive experiments to evaluate whether LEAST module is necessary for Deep RL."? Introducing a trainings heuristic can be helpful, but I do not understand what you mean by necessary. Please elaborate.

Can you summarize comprehensibly why the proposed criteria of the LEAST heuristic is a good choice in general for stopping episodes? The experimental evidence seems not entirely clear. Especially why the proposed dual-criteria would be better than possible multi-criteria.

**Relation To Broader Scientific Literature:**

The paper contributes to the online RL literature. Due to its nature to terminate episodes early, potentially saving computation costs, it might be relevant to computationally heavy experiments.

**Theoretical Claims:**

Theoretical foundation for the presented heuristic would be nice, but is not included in the paper.

---

> ### Author Rebuttal · Authors · 2025-04-01
>
> Thanks for the insightful suggestions and constructive feedback on our work. Please find our responses to each of the concerns below.
>
> **Q1: What do you mean in Background by "In this paper, we investigate whether deep RL agents deployed on cumulative reward maximization tasks also suffer from sunk cost fallacy."?**
>
> We investigate whether RL agents exhibit behavior analogous to the sunk cost fallacy in reward maximization tasks, where agents inefficiently persist in executing trajectories despite diminishing returns will cause more losses, similar to how investors might hold onto losing investments. Through this analysis, we aim to identify and address fundamental inefficiencies in conventional RL architectures.
>
> **Q2: What makes the introduced heuristic special for deep RL in comparison to its importance for "non-deep" RL?**
>
> We would like to clarify that our proposed methodology is general for RL (not specific to Deep RL).
>
> We validate our method's generality through experiments on tabular Q-learning in two MiniGrid navigation tasks [1] with default parameters. Results below measure the episode number required to converge to the optimal policy. LEAST significantly improves learning efficiency compared to standard Q-learning (**5%-14%** reduction in episodes required to converge), demonstrating its effectiveness in improving learning efficiency in simpler, discrete non-deep RL settings.
>
> | Method | 6x6 with obstables ($\downarrow$) | 8x8 with obstables ($\downarrow$) |
> | --- | --- | --- |
> | **Q learning w/ LEAST** | $428.0\pm21.3$ | $495.3\pm24.7$ |
> | Q learning | $494.7\pm28.9$ | $523.3\pm26.3$ |
>
> **Why we mainly discuss deep RL:** While non-deep RL provides a valuable testbed for validating the algorithms clearly,  our main focus is to make deep RL algorithms learn better and use data more efficiently, since Deep RL is critical for solving complex, high-dimensional problems and has unique challenges and practical relevance in real-world applications. We aim to validate our approach in these demanding scenarios while encouraging future theoretical analysis in simpler RL paradigms.
>
> We hope our work can inspire exploration in non-deep RL, e.g. tabular & linear methods for theoretical analysis.
>
> [1] Open source GitHub repo: Minigrid-RL (default setting)
>
> **Q3: What do you mean by "In this section, we conduct comprehensive experiments to evaluate whether LEAST module is necessary for Deep RL."?**
>
> We will revise the sentence for clarity in the revised version. To clarify, in this section, we aim to evaluate the necessity of LEAST by investigating whether it provides significant improvements in the performance & efficiency of baselines.
>
> **Q4: ... why the proposed criteria of the LEAST are a good choice in general? ... Especially why is this better than a possible multi-criteria.**
>
> The proposed dual-criteria based on Q-values & loss offer the following major advantages, making it a robust, versatile, and general choice for our LEAST framework:
>
> - **Robustness**. Q-values assess trajectory quality through expected rewards, while critic loss indicates the agent's understanding (familiarity) of transitions. These complementary metrics provide comprehensive evaluation criteria for stopping decisions.
> - **Versatility**. Q-values & loss are core metrics in RL algorithms, making them algorithm-agnostic & environment-independent, unlike entropy (SAC-specific) or scene-specific metrics. This enables the dual-criteria to work across different tasks and algorithms with minimal changes.
> - **Simplicity**. Q-values & loss are readily computed metrics that require no extra computation (e.g., network modules), making the criteria efficient to implement.
>
> We explored various multi-criteria but found none that consistently outperformed the dual-criteria in efficiency and task independence. Our motivation is to provide a fundamental framework that is simple, effective, and broadly applicable across diverse tasks and algorithms, and to encourage future research into more sophisticated multi-criteria approaches.
>
> **W1: The presented background has nothing to do with "Deep" RL, it describes RL basics. It seems reasonable...**
>
> We revised this paragraph in the background section to better reflect the context of deep RL.
>
> **C1: The connection between the observed problem & the sunk-cost fallacy seems far-fetched...**
>
> Our analogy here focuses on the behavioral similarity: the implicit commitment to completing trajectories without evaluating their future benefit, although the classical sunk-cost fallacy typically involves emotional attachment. The absence of early stopping mechanisms in existing RL architectures leads to decision-making inertia, resembling the sunk-cost fallacy in its effect on efficiency.
>
> ---
> Thank you again for reviewing our work. We will refine the typos and hope our responses address the concerns and would appreciate your consideration in the evaluation. We are happy to provide further clarification if needed.

---

> > ### Comment · Reviewer_Rcz7 · 2025-04-07
> >
> > Thank you for your comprehensive rebuttal and for addressing the points raised in my initial review.
> > I still think sunk-cost fallacy is not a good fit to the introduced method. Nonetheless early stopping is a valuable mechanism in the presented context.
> > Having read the rebuttal as well as the other reviews, I consider raising my score from 2 to 3. And thus, leaning more towards the acceptance of the paper.

---

> > > ### Author Response · Authors · 2025-04-08
> > >
> > > We sincerely thank the reviewer for the follow-up comments and for reconsidering your initial assessment of our work. We appreciate your thoughtful engagement and recognition of early stopping as a valuable mechanism in the presented context.
> > >
> > > Regarding the analogy to the sunk cost fallacy, our intention was not to equate the two concepts directly (as the classic definition involves emotional attachment -- which is absent in RL as agents typically do not have emotions), but rather to highlight the **behavioral similarity**: the implicit commitment to completing trajectories without dynamically evaluating their future benefit. This **decision-making inertia**, stemming from the absence of early stopping mechanisms, produces inefficiencies similar to those observed in the sunk cost fallacy. We aim to demonstrate how this analogy highlights the inefficiencies in existing RL frameworks and the relevance of our proposed method as a solution.
> > >
> > > *We will revise the manuscript to better clarify this point and refine the explanation in the introduction to better illustrate this connection within the context of this paper:*
> > > > Existing RL architectures often lack a mechanism to dynamically evaluate the potential utility of continuing a trajectory. This results in agents persistently executing full trajectories, even when it becomes clear that further interactions are unlikely to yield meaningful outcomes and new insights.
> > >
> > > >This decision-making inertia reflects a behavioral inefficiency akin to the sunk cost fallacy, where individuals persist in a course of action simply because resources have already been invested, without reassessing whether continuing these actions is worthwhile. However, unlike humans, who can reassess and decide to stop when recognizing a clear loss, RL agents are constrained by existing frameworks that focus exclusively on policy optimization without mechanisms to evaluate and interrupt unproductive trajectories.
> > >
> > > >While the sunk cost fallacy in humans is traditionally linked to emotional attachment, this analogy is drawn in this context to highlight the structural inefficiency in RL caused by the absence of dynamic stopping mechanisms. Specifically, RL agents do not have the ability to evaluate the declining utility of continuing a trajectory, leading to wasted computational resources and lower learning efficiency. LEAST addresses this critical issue by introducing a dynamic stopping mechanism that enables agents to identify and terminate low-quality and redundant trajectories early, thereby reallocating resources to more promising ones. This not only mitigates inefficiencies reminiscent of the sunk cost fallacy but also improves sample efficiency and overall learning performance.
> > >
> > > In addition, to further contextualize this analogy, we will expand the background section to include real-world examples, such as continued investment in failing stocks. These examples demonstrate behavioral parallels between human decision-making related to sunk cost fallacy and the interacting behavior of RL agents, where the lack of dynamic stopping mechanisms leads to persistent yet inefficient trajectory execution. By providing this refined explanation and including real-world examples, we aim to better illustrate the behavioral parallels and highlight the practical relevance of our proposed method.
> > >
> > > ----
> > >
> > > *Lastly, as you kindly mentioned the possibility of raising your score, we would be grateful if this could be reflected in your final evaluation (edit the review). We also welcome any further suggestions you may have and will definitely incorporate them into the revision.*
> > >
> > > Your thoughtful reconsideration and feedback have already contributed significantly to improving the quality and clarity of our work, and we deeply value your support in the review process.

---

### Official Review · Reviewer_zKrJ · 2025-03-12

**Overall Recommendation:** 4

**Summary:**

The paper proposes a novel technique for improving RL training that allows an agent to terminate an episode early if the expected return drops below a heuristic threshold. The paper aims to show that this can speed up training as it avoids filling the replay buffer with uninformative trajectories. The authors provide evidence for their claim across a variety of benchmarks and baseline algorithms.


### Update after rebuttal period

With additional context, I support the acceptance of this paper. All major questions were adequately addressed.

**Claims And Evidence:**

The paper is empirical in nature and provides a variety of evidence for its claims. The evidence seems sufficient for the claims.

**Essential References Not Discussed:**

Relevant prior work from the field of robotics seems to not be discussed, such as [1] and follow up work. As robotics is a field where limiting long episodes has strong practical relevance, this seems important.

[1] Leave no Trace: Learning to Reset for Safe and Autonomous Reinforcement Learning, Eysenbach et al, ICLR 2018 https://openreview.net/forum?id=S1vuO-bCW

**Experimental Designs Or Analyses:**

Aside from the concerns raised above, the experimental design is well done.

**Methods And Evaluation Criteria:**

The bench-marking covers several different RL problems and algorithms. A small (rather nitpicky) caveat is that the baseline RL algorithms use are relatively old and have known weaknesses such as problems with learning stability. All considered algorithms are 4 years old at this point. To address this, I would encourage the authors to verify their findings with more up-to-date architectures and algorithms such as SR-SAC [1], BRO [2], Simba [3], CrossQ [4], TD-MPC2 [5], or MAD-TD [6]. Obviously I am not asking the authors to provide evidence on all of these, but for example repeating the experiments on Mujoco tasks with a model-free algorithm shouldn't be too much of an ask. Model-based algorithms such as TD-MPC2, Dreamer or MAD-TD might be another interesting direction, as the gathered data influences both model and value function learning. In the image based domains, why wasn't LEAST evaluated with TACO or A-LIX as well? The improvements should be orthogonal?

One additional question for clarification: in the reported scores, were the algorithms evaluated with or without early resets? I think it would be important to verify that the additional performance is not due to the algorithms being able to avoid "difficult" parts of the state space they might naturally have to end up in. Clarifying this is a requirement for me supporting the acceptance of the paper!

[1] Sample-Efficient Reinforcement Learning by Breaking the Replay Ratio Barrier, D'Oro et al, ICLR 2023 https://openreview.net/forum?id=OpC-9aBBVJe

[2] Bigger, Regularized, Optimistic: scaling for compute and sample-efficient continuous control, Nauman et al, NeurIPS 2024, https://openreview.net/forum?id=fu0xdh4aEJ&noteId=It2lExTd92

[3] SimBa: Simplicity Bias for Scaling Up Parameters in Deep Reinforcement Learning, Lee et al, ICLR 2025, https://openreview.net/forum?id=jXLiDKsuDo

[4] CrossQ: Batch Normalization in Deep Reinforcement Learning for Greater Sample Efficiency and Simplicity, Bhatt et al, ICLR 2024, https://openreview.net/forum?id=PczQtTsTIX

[5] TD-MPC2: Scalable, Robust World Models for Continuous Control, Hansen et al, ICLR 2024, https://openreview.net/forum?id=Oxh5CstDJU

[6] MAD-TD: Model-Augmented Data stabilizes High Update Ratio RL, Voelcker et al., ICLR 2025, https://openreview.net/forum?id=6RtRsg8ZV1

**Other Comments Or Suggestions:**

none

**Other Strengths And Weaknesses:**

I have two problems with the presentation of the paper, but these are relatively minor. First of all, the algorithm is presented as the agent "learning" to avoid the sunk cost fallacy. However, LEAST is a fixed heuristic, so no learning is happening as far as I can tell. In addition, the sunk cost fallacy seems to be only somewhat applicable here, as agents do not normally have a choice but to continue in a trajectory until the end.

The other problem is that the authors mostly consider environments which already include early resets, such as the mujoco tasks. In these, avoiding "bad" areas of the state space is indeed optimal, as the agent is most likely facing an early reset anyways. However, in other tasks with less well shaped rewards and early reset structure, the method might limit exploration to a problematic extend. I would ask the authors to comment on this more clearly. The authors already comment on some exploration related problems in 3.2, but I believe that the chosen test tasks might lead to optimistic conclusions on the methods peformance.

**Questions For Authors:**

See above

**Relation To Broader Scientific Literature:**

They seem to fit the literature.

**Theoretical Claims:**

No theoretical claims were made.

---

> ### Author Rebuttal · Authors · 2025-04-01
>
> We thank the reviewer for the insightful suggestions and useful feedback of our work. Please find our responses to each of the concerns below.
>
> ## Question
>
> **Q1.1: Please verify the findings with more up-to-date algorithms such as SR-SAC, CrossQ,...,e.g., repeating the experiments on Mujoco tasks with a model-free algorithm.**
>
> Thanks for your suggestion. We have incorporated CrossQ as an additional backbone algorithm and conducted experiments on Hopper & HalfCheetah (with full results to be included in the revision due to time constraints).
>
> The results show that CrossQ-LEAST consistently outperforms vanilla CrossQ in both performance & learning efficiency. This improvement may stem from CrossQ's more stable Q-value predictions compared to SAC, which helps calculate more reliable stop thresholds.
>
> Table: Score$\uparrow$ (Convergence steps (M)$\downarrow$ )  (Average of 3 runs):
>
> | Method | Hopper| HalfCheetah |
> | --- | --- | --- |
> | **CrossQ-LEAST** | $2853\pm162.37$ ($3.25$) | $13607.57\pm 115.48$ ($2.26$) |
> | CrossQ | $1839.36\pm297.24$ ($4.71$) | $13035.81\pm 137.16$ ($2.67$) |
>
> **Q1.2: Why wasn't LEAST evaluated with TACO or A-LIX as well? The improvements should be orthogonal?**
>
> Since both methods are based on DrQv2, we used it as our primary backbone to evaluate LEAST's general improvements. We also tested LEAST with A-LIX and found consistent performance gains across image-based domains (results to be included in revision). Besides, MBRL indeed remains an interesting direction for future work.
>
> | Method | Finger Turn Hard | Quadruped Run |
> | --- | --- | --- |
> | **A-LIX w/ LEAST** | $537\pm19.24$ | $858.73\pm 16.81$ |
> | A-LIX | $451\pm32.61$  | $772.53\pm 25.97$ |
>
> **Q2: Relevant prior work from the field of robotics seems not to be discussed.**
>
> We appreciate this suggestion and will enhance the related work section in the revision to thoroughly discuss relevant research from the robotics field.
>
> ## Weakness
>
> **W1.1: The algorithm is presented as the agent "learning" to avoid the sunk cost fallacy. But no learning is happening as far as I can tell.**
>
> We appreciate your suggestion and will revise the phrasing to use "decide" instead of "learn," as it more accurately reflects how LEAST functions. We originally used the learning analogy to illustrate the behavioral patterns that LEAST promotes.
>
> **W1.2: The sunk cost fallacy seems to be only somewhat applicable here...**
>
> Our goal is to make it more intuitive for readers to understand our core contribution: existing RL architectures often lack the capability to dynamically evaluate the potential utility of continuing a trajectory, leading agents to "stubbornly" execute the full trajectory regardless of its relevance or productivity within the given interaction budget. This inefficiency bears an analogy to the sunk cost fallacy, where agents implicitly persist in a trajectory simply because it has already been initiated, without reassessing whether it remains promising or worthwhile. While the sunk cost fallacy is traditionally associated with human decision-making, we draw this analogy to highlight how LEAST's dynamic stopping mechanism addresses a similar inefficiency in RL.
>
> **W2: In other tasks with less well-shaped rewards & early reset structure, the method might limit exploration to a problematic extent?**
>
> LEAST's adaptive stopping mechanism functions independently of environment-level reinitialization, as it can detect suboptimal states before the environment triggers a reset. In locomotion tasks, e.g., LEAST identifies patterns like in the agent instability that indicate imminent failure, proactively stopping the episode before environment-level reinitialization occurs—particularly during early training stages. This proactive stopping enables LEAST to leverage its dynamic stopping mechanism to improve learning efficiency, even in environments with early reinitialization (as demonstrated in Figs 6, 7, 9).
>
> To address the exploration limitations, we introduce a simple yet effective method, dynamic exploration noise adjustment, and find that it effectively helps agents escape suboptimal trajectories to explore more diverse behaviors. Thus, in environments with less well-shaped rewards & early reinitialization structure (e.g., PointMaze), LEAST still demonstrates robust performance, further confirming the method’s general applicability.
>
> *We hope our work can inspire future research to build upon this foundation for more challenging scenarios. Exploring further improvements to LEAST's exploration capabilities remains an exciting direction for future work.*
>
> ---
>
> We would like to thank the reviewer once again for the time and effort in reviewing our work! We are happy to provide further clarification if you have any additional questions. We hope that our responses adequately address your concerns, and we would greatly appreciate it if you could kindly consider reflecting on this in the evaluation.

---

> > ### Comment · Reviewer_zKrJ · 2025-04-01
> >
> > Hi, I read the rebuttal, thanks for testing the additional algorithmic setups!
> >
> > I briefly wanted to point your attention at this question:
> >
> > "One additional question for clarification: in the reported scores, were the algorithms evaluated with or without early resets? I think it would be important to verify that the additional performance is not due to the algorithms being able to avoid "difficult" parts of the state space they might naturally have to end up in. Clarifying this is a requirement for me supporting the acceptance of the paper!"
> >
> > As I said, this is a clarification, but I think it is vital that this is correctly handled.

---

> > > ### Author Response · Authors · 2025-04-02
> > >
> > > > **Question:** Clarifying this is a requirement for me supporting the acceptance of the paper: in the reported scores, were the algorithms evaluated with or without early resets?.
> > >
> > > We sincerely apologize for mistakenly combining this question with a subsequent one in our initial rebuttal due to space limitations, which caused us to overlook addressing it fully. We greatly appreciate the reviewer for raising this important question again, for the patience, and for giving us the opportunity to address it thoroughly.
> > >
> > > To clarify, all algorithms in our experiments were **evaluated** **without** **early resets**, ensuring a fair and rigorous comparison. This experimental setup guarantees that the reported performance improvements are not the result of artificially "avoiding difficult parts of the state space," but rather reflect the genuine gains in learning stability, efficiency, and adaptability achieved by our method. To ensure complete transparency, we will explicitly highlight this point in the revised manuscript. Furthermore, we will include this clarification in the README file accompanying our codebase, which will be open-sourced upon publication, to ensure that this aspect is fully documented.
> > >
> > > We would like to thank the reviewer for acknowledging our additional experiments and explanations, and for careful consideration of our work and the emphasis on this critical aspect. Please let us know if you have any further questions or if additional clarifications are needed, and we are more than happy to provide further details.

---

### Official Review · Reviewer_jmK5 · 2025-03-15

**Overall Recommendation:** 3

**Summary:**

This paper introduces "Learn to Stop" (LEAST) to address the "sunk cost fallacy" in deep reinforcement learning (RL). The sunk cost fallacy refers to how RL agents must complete episodes even if the trajectory collected thus far is already poor, which ultimately provides low-quality data to the agent. LEAST allows the agent to terminate episodes early by dynamically evaluating the quality and learning potential of current trajectories based on Q-values and critic gradient information. Empirically, LEAST consistently improves sample efficiency and final performance compared to baseline methods.

**Claims And Evidence:**

**Claim 1: The traditional RL framework suffers from the sunk cost fallacy, leading to inefficient sampling and suboptimal policy learning.**

  * **Unsupported.** The core experiments do not show that LEAST is truncating suboptimal trajectories but instead focus on data efficiency improvements. The Maze examples in Figure 3 shows that LEAST agents have a higher fraction of high-return trajectories, but again does not show that LEAST is truncating suboptimal trajectories.

**Claim 2: LEAST improves sample efficiency and final performance across different algorithms**
  * **Supported.** Figures 6 and 9 show that algorithms enhanced with LEAST require fewer steps to reach the same performance level as their vanilla counterparts.

**Essential References Not Discussed:**

None.

**Experimental Designs Or Analyses:**

1. **Figure 3 is difficult to interpret.** Could you clarify what "Loss" refers to specifically? Is it the critic's loss, TD error, or the policy's return? Additionally, what criteria determine whether a sample is classified as "High/Low Loss" versus "High/Low Q"?
1. Why does Figure 5 only present results for Ant and HalfCheetah? Are the results for other environments less compelling? If so, what might explain this discrepancy?
1.  **The paper lacks empirical evidence demonstrating that LEAST actually terminates low-return trajectories early.** This verification is essential since early termination of unpromising trajectories is the fundamental motivation behind the algorithm.
1. **The plasticity experiments seem disconnected from the main contribution.** The paper's shift to studying network plasticity appears abrupt and inadequately motivated, and these experiments feel preliminary. While Figure 11 suggests LEAST reduces plasticity loss rates, this observation may be misleading. One of LEAST's primary motivations is that completing obviously low-quality episodes provides poor training data, potentially leading to premature convergence to suboptimal policies. Would this premature convergence potentially lead to faster loss of plasticity?

**Methods And Evaluation Criteria:**

1. The evaluation criteria and benchmarks (MuJoCo and DMC) are standard and appropriate for the problem. The authors also make a good case for why these environments are suitable for testing their method, particularly in environments with potential dead-ends or suboptimal trajectories.

1. What do the edges of the box in the box plots represent?

**Other Comments Or Suggestions:**

1. "which proves that it is trapped in sub-optimality and can’t escape." While I would agree that the Vanilla training curve for the Medium maze hints at suboptimal convergence, it is unclear if the Vanilla training curve for the Large maze has converged yet. "Prove" seems like too strong of a word here. Plotting the entropy of the policy throughout training could make the argument for suboptimal convergence stronger.
1. "Figure 4 shows this dual-criteria threshold (ωi × εi ) significantly outperforms purely Q-based threshold" What does "significantly" mean here, precisely? Figure 4 shows that the 25th percentile dual criteria threshold agents outperform the 75th percentile Q-based threshold agents, but does this constitute a significance test? (What do the edges of the box in each box plot represent? I assumed 25/75th percentile). This statement should be rephrased to explicitly state what metrics we are comparing.
1. "Makes it difficult to measure central tendency." is a sentence fragment.
1. "Specifically, for TD3 and SAC (Figure 6(a,b))." is a sentence fragment
1. Figure 7 should have error bars
2. Line 327: "mwthod" -> method
3. line 328: "thebox" -> the box
4 "thus avoiding the agent from training via" -> thus preventing the agent from training on
5. Line 385: "sequential of skills" it is unclear what this means
6. Line 385: "Compare to TD3" -> compared
7. "Appendix D.2 detailed introduces" -> extra word
1. "We analyze the impact of the startup time of LEAST in Appendix D.4, it is a good choice to start LEAST from 10% − 20% of the total training time for MuJoCo." is a run-on sentence
1. "for the image input task" -> tasks
1. Line 643: broken reference to DDPG
1. Line 796: "Deep RL" -> Deep RL
1. Line 932: "noisy daily"
1. "Divide quadrants uniformly using the mean of Q and Loss of advanced agent buffer samples." is a sentence fragment
1. Line 797: Broken reference for REDQ.

**Other Strengths And Weaknesses:**

**Additional Strengths:**

1. LEAST is intuitive and easy to understand.
2. The empirical validation is comprehensive.

**Additional Weaknesses:**

1. While the basic Q-based threshold version of LEAST presents a sound approach, the additional mechanisms appear to function more as heuristic patches rather than integrated components of the algorithm's core design.
1. The paper would benefit from exploring potential tradeoffs of early stopping, particularly how it might affect the discovery of valuable long-term strategies that only become apparent after extended training periods.
1. The paper contains several typos and sentence fragments that occasionally impact readability (specific examples are provided in the next section).

**Author comments on these weaknesses would be appreciated, though I emphasize that these weaknesses are not critical to my evaluation.**

**Questions For Authors:**

**I currently lean to reject. While experiments demonstrate LEAST's improved data efficiency, the paper lacks convincing evidence that this improvement stems from truncating low-quality trajectories as claimed.**

1. Can the authors provide additional experiments showing that LEAST is indeed truncated low-quality trajectories? For instance, one could plot the distribution of trajectory lengths thoughout training for Vanilla and LEAST agents (in some fair manner).

2. Could the authors contextualize the plasticity loss experiments more? My impression is that the plasticity loss experiments would be more appropriately placed in an appendix, as their relevance to the core contributions seems limited. This would free up space for more informative ablation studies that could better illuminate the algorithm's effectiveness.

**If the authors address both of these points, I will consider raising my score.**

**Relation To Broader Scientific Literature:**

The paper appropriately positions its contribution within the RL literature, particularly in relation to sample efficiency methods and existing approaches to early stopping. The authors discuss related work in Appendix A, covering both sample efficiency techniques in Deep RL and early stopping mechanisms.

**Theoretical Claims:**

None.

---

> ### Author Rebuttal · Authors · 2025-04-01
>
> We thank the reviewer for the insightful suggestions and useful feedback of our work. Please find our responses to each of the concerns below.
>
> **Q1: Can you demonstrate experimentally that LEAST truncates low-quality trajectories?**
>
> We conduct additional experiments on Ant to analyze trajectory length and quality, and will include it in the revision. Experiments below confirm that LEAST effectively truncates low-quality trajectories.
>
> **LEAST effectively truncates trajectories:** We compare the distribution of trajectory lengths stored in the replay buffer between LEAST and vanilla agents (built upon TD3) during different stages of training. Table 1a shows that LEAST stores significantly more short trajectories than the vanilla agent, especially during the early training stages.
>
> Table 1a. Percentage distribution of lengths in the buffer at different training stages (25%, 50%, 75%, and 100%). Trajectory length (Short (S): $l \leq 333$, Medium (M): $333<l \leq 666$, Long (L): $l > 666$):
>
> | Method | 25% Progress | 50% Progress | 75% Progress | 100% Progress |
> | --- | --- | --- | --- | --- |
> | **LEAST-TD3** | S: 27.25, M: 46.14, L: 24.61 | S: 45.85, M: 38.62, L: 15.53 | S: 23.75, M: 39.17, L: 37.11 | S: 4.42, M: 6.35, L: 89.23 |
> | Vanilla-TD3 | S: 21.17, M: 52.38, L: 26.45 | S: 8.69, M: 15.03, L: 76.28 | S: 4.47, M: 8.24, L: 87.29 | S: 1.18, M: 7.51, L: 91.31 |
>
> **Truncated trajectories are low-quality.**  We design a comparison experiment that evaluates the quality of trajectories identified for truncation by LEAST. We periodically (every 10k steps) stop the truncation to allow these "would-be truncated" trajectories to execute fully until their natural termination. The final cumulative rewards of these trajectories are then compared to the average rewards of trajectories under normal LEAST training. Results below show that "would-be truncated" trajectories achieve lower rewards across all training phases, confirming that LEAST effectively truncates suboptimal trajectories. This is further supported by suboptimal path truncation visualizations in MAZE ([ANONYMOUS LINK](https://anonymous.4open.science/r/lzvk-EF05/R_1.png)).
>
> Table 1b. Effectiveness of LEAST in truncating low-quality trajs:
>
> | Method | 25% Progress | 50% Progress | 75% Progress | 100% Progress |
> | --- | --- | --- | --- | --- |
> | **would-be-truncated** | $1619.26\pm1033.58$ | $3361.37\pm653.27$ | $4783.07\pm468.63$ | $4973.48\pm502.86$ |
> | **Baseline** | $3975.26\pm507.91$ | $4873\pm282.35$ | $5523\pm361.24$ | $6376\pm417.78$ |
>
> **Q2: The plasticity loss experiments would be more appropriately placed in an appendix...**
>
> We will move this to the appendix and use the space to expand the analyses in appendix.
>
> The plasticity loss experiments were included to investigate a secondary benefit of LEAST in mitigating plasticity loss. Premature convergence caused by low-quality data (primacy bias [1]) is correlated with plasticity loss [2]. Since LEAST improves buffer data quality by truncating suboptimal trajectories (Fig. 3), we hypothesized it could mitigate plasticity loss, as analyzed in Fig. 11 to provide preliminary evidence for future work.
>
> [1] The primacy bias in deep reinforcement learning. ICML'22.
>
> [2] Loss of Plasticity in Continual Deep Learning. Nature'24.
>
> **W1: The additional mechanisms appear to function more as heuristic patches...**
>
> These auxiliary mechanisms ensure LEAST's robustness in complex environments.
>
> While the basic Q-based module works well on simple tasks like Hopper, but becomes less reliable in more complex environments, e.g. Ant. We propose two complementary modules: a dynamic buffer to improve threshold calculations by filtering outliers, and noise adjustments to maintain exploration despite early stopping.
>
> **W2: The paper would benefit from exploring potential tradeoffs of early stopping and exploration...**
>
> To achieve that tradeoff, we introduce a simple yet effective noise adjustment module (which reduces the frequency of early stopping), helping to maintain exploration diversity and mitigate the risk of converging to overly short-term strategies. We hope our work can inspire research for further improvement.
>
> **W3: Clarify what "Loss" refers to specifically in Fig 3?**
>
> Loss refers to critic loss. We classify samples as High/Low Q based on their Q-values compared to the median Q-value in the Q buffer (and similarly for High/Low loss).
>
> **W4: Fig 5 only presents results for Ant and HC.**
>
> We chose these tasks as they represent different complexity levels - Ant: complex; HC: simpler - allowing us to evaluate LEAST across varying challenges. We will include full results in the revision.
>
> **W5: Typos, fragments & box plot.**
>
> We improved them in the revision. The edge of box plots indicates upper and lower bounds.
>
> ---
> Thanks again for reviewing our work! We hope our responses address the concerns and would appreciate your consideration in the evaluation. We are happy to provide further clarification if needed.

---

> > ### Comment · Reviewer_jmK5 · 2025-04-07
> >
> > Thank you for your detail response! In particular, I appreciate the additional results supported that LEAST truncated low-quality trajectories. These results fill an important gap currently in the paper, and I do urge the authors to include them in revisions (either in a camera-ready or resubmission). Assuming the plasticity experiments will be moved to the appendix (and the appendix makes it clear that these are fairly preliminary results) and the typos/fragments will be addressed, I am raising my score.
> >
> > Overall, it's a cool paper.

---

> > > ### Author Response · Authors · 2025-04-08
> > >
> > > We sincerely thank Reviewer jmK5 for the thoughtful feedback and the time and effort in evaluating our work, and we deeply appreciate your recognition of the contributions of our work.
> > >
> > > We will carefully address all your suggestions in the revised manuscript (including incorporating additional quantitative and qualitative results on truncating low-quality trajectories into the main text, relocating the plasticity experiment to the appendix, and correcting typos and fragments), and these suggestions have been invaluable in refining the quality, clarity, and overall impact of our work.

---

### Decision · Program_Chairs · 2025-05-01

**Decision:**

Accept (poster)

**Comment:**

This paper introduces LEAST, a mechanism designed to improve sample efficiency in deep reinforcement learning by allowing agents to terminate unproductive episodes early. The core idea is framed as overcoming the "sunk cost fallacy," where agents continue episodes even when they are unlikely to yield useful data, thus polluting the replay buffer. LEAST uses Q-values and critic loss statistics to decide when to stop an episode prematurely.

Initial reviews were mixed but generally acknowledged the potential value of the proposed method. Reviewers appreciated the simplicity and intuition behind LEAST, along with its empirical validation across various benchmarks and algorithms. However, several concerns were raised. Reviewer jmK5 questioned whether the method demonstrably truncates low-quality trajectories as claimed and found the experiments on network plasticity somewhat disconnected from the main contribution. Reviewer zKrJ, while positive, pointed out the use of older baseline algorithms and questioned the aptness of the "sunk cost fallacy" terminology, also requesting clarification on the evaluation protocol regarding early resets. Reviewer Rcz7 also found the sunk cost analogy far-fetched and questioned the general applicability and necessity of the specific heuristic criteria used by LEAST. Reviewer 4RG4 raised concerns about the potential unreliability of using training loss as a criterion and the somewhat ad-hoc justification for components like the dynamic buffer and exploration noise adjustment.

The authors provided a detailed rebuttal, addressing the key concerns effectively. They presented additional experiments specifically demonstrating that LEAST does indeed truncate shorter, lower-quality trajectories, satisfying Reviewer jmK5's main reservation. They agreed to move the plasticity experiments to the appendix. In response to Reviewer zKrJ, they showed positive results incorporating LEAST with newer algorithms like CrossQ and A-LIX and crucially clarified that evaluations were performed without allowing LEAST agents to benefit from early resets during the evaluation phase, ensuring a fair comparison. They defended the choice of dual-criteria (Q-value and loss) for its generality and simplicity, while also demonstrating LEAST's applicability in non-deep RL settings for Reviewer Rcz7. While acknowledging the "sunk cost fallacy" analogy might not be a perfect fit (lacking the emotional component), they argued for its intuitive value in conveying the problem of inflexible trajectory execution and agreed to refine the explanation. Concerns from Reviewer 4RG4 regarding loss unreliability and the motivation for auxiliary mechanisms were also addressed with further explanations.

Following the rebuttal, all reviewers acknowledged the authors' responses. Reviewer jmK5 and Reviewer Rcz7 explicitly raised their scores to Weak Accept. Reviewer zKrJ and Reviewer 4RG4 maintained their Accept recommendations, expressing satisfaction with the clarifications. The consensus among reviewers shifted positively, recognizing the paper's contribution in addressing an inefficiency in DRL sample collection with a novel and practical approach, supported by strong empirical evidence after the rebuttal.